# Asynchronous Training Schemes in Distributed Learning with Time Delay

**Haoxiang Wang**                                                                          *whx22@mails.tsinghua.edu.cn*
*Department of Automation*
*Tsinghua University*

**Zhanhong Jiang**                                                                                    *zhjiang@iastate.edu*
*Translational AI Center*
*Iowa State University*

**Chao Liu**                                                                                        *cliu5@tsinghua.edu.cn*
*Department of Energy and Power Engineering*
*Tsinghua University*

**Soumik Sarkar**                                                                                  *soumiks@iastate.edu*
*Department of Mechanical Engineering*
*Iowa State University*

**Dongxiang Jiang**                                                                            *jiangdx@tsinghua.edu.cn*
*Department of Energy and Power Engineering*
*Tsinghua University*

**Young M. Lee**                                                                                     *young.m.lee@jci.com*
*Johnson Controls*

**Reviewed on OpenReview:** *https://openreview.net/forum?id=z0GJxw07Z6*

## Abstract

In the context of distributed deep learning, the issue of stale weights or gradients could result in poor algorithmic performance. This issue is usually tackled by delay tolerant algorithms with some mild assumptions on the objective functions and step sizes. In this paper, we propose a different approach to develop a new algorithm, called **P**redicting **C**lipping **A**synchronous **S**tochastic **G**radient **D**escent (aka, PC-ASGD). Specifically, PC-ASGD has two steps - the *predicting step* leverages the gradient prediction using Taylor expansion to reduce the staleness of the outdated weights while the *clipping step* selectively drops the outdated weights to alleviate their negative effects. A tradeoff parameter is introduced to balance the effects between these two steps. Theoretically, we present the convergence rate considering the effects of delay of the proposed algorithm with constant step size when the smooth objective functions are weakly strongly-convex, general convex, and nonconvex. One practical variant of PC-ASGD is also proposed by adopting a condition to help with the determination of the tradeoff parameter. For empirical validation, we demonstrate the performance of the algorithm with four deep neural network architectures on three benchmark datasets.

## 1 Introduction

The availability of large datasets and powerful computing led to the emergence of deep learning that is revolutionizing many application sectors from the internet industry and healthcare to transportation and energy Gijzen (2013); Wiedemann et al. (2019); Gao et al. (2022); Liu & Liu (2023). As the applications are

scaling up, the learning process of large deep learning models is looking to leverage emerging resources such as edge computing and distributed data centers privacy preserving. In this regard, distributed deep learning algorithms are being explored by the community that leverage synchronous and asynchronous computations with multiple computing agents that exchange information over communication networks Lian et al. (2017); Cao et al. (2023); Qian et al. (2022). We consider an example setting involving an industrial IoT framework where the data is geographically distributed as well as the computing resources. While the computing resources within a local cluster can operate in a (loosely) synchronous manner, multiple (geographically distributed) clusters may need to operate in an asynchronous manner. Furthermore, communications among the computing resources may not be reliable and prone to delay and loss of information.

The master-slave and peer-to-peer are two categories of distributed learning architectures. On one hand, Federated Averaging and its variants are considered to be the state-of-the-art for training deep learning models with data distributed among the edge computing resources such as smart phones and idle computers Hard et al. (2018); Sattler et al. (2019). PySyft Ryffel et al. (2018) and its robust version Deng et al. (2020), the scalable distributed DNN training algorithms Strom (2015) and more recent distributed SVRG Cen et al. (2020) and clustered FL Sattler et al. (2021) are examples of the master-slave architecture. On the other hand, examples of the peer-to-peer architecture include the gossip algorithms Blot et al. (2016); Even et al. (2020); Li et al. (2021); Tu et al. (2022), and the collaborative learning frameworks Jiang et al. (2017); Liu et al. (2019).

However, as mentioned earlier, communication delay remains a critical challenge for achieving convergence in an asynchronous learning setting Chen et al. (2016); Tsianos et al. (2012) and affects the performances of the frameworks above. Furthermore, the amount of delay could be varying widely due to artifacts of wireless communication and different devices. To eliminate the negative impact of varying delays on the convergence characteristics of distributed learning algorithms, this work proposes a novel algorithm, called **P**redicting **C**lipping **A**synchronous **S**tochastic **G**radient **D**escent (aka, PC-ASGD). The goal is to solve the distributed learning problems involving multiple computing or edge devices such as GPUs and CPUs with varying communication delays among them. Different from traditional distributed learning scenarios where synchronous and asynchronous algorithms are considered separately, we take both into account together in a networked setting.

Table 1: Comparisons between asynchronous algorithms

| Methods | $f$ | $\nabla f$ | Delay Ass. | Con.Rate | D.C. | G.C. | A.S. |
|---|---|---|---|---|---|---|---|
| ASGD Dean et al. (2013) | Non-convex | Lip. | Bou. | $\mathcal{O}(\frac{1}{\sqrt{T}})$ | ✗ | ✗ | ✗ |
| DC-ASGD Zheng et al. (2017) | Str-con | Lip. | Bou. | $\mathcal{O}(\frac{1}{T})$ | ✗ | ✓ | ✗ |
| | Non-convex | Lip. | Bou. | $\mathcal{O}(\frac{1}{\sqrt{T}})$ | ✗ | ✓ | ✗ |
| D-ASGD Lian et al. (2017) | Non-convex | Lip.&Bou. | Bou. | $\mathcal{O}(\frac{1}{\sqrt{T}})$ | ✓ | ✗ | ✗ |
| DC-s3dg Rigazzi (2019) | Non-convex | Lip. | Unbou. | N/A | ✓ | ✓ | ✗ |
| AGP Assran & Rabbat (2020) | Str-con | Lip. | Bou. | $\mathcal{O}(\frac{1}{T} + \frac{1}{T^\zeta} + \frac{1}{T^{1-\zeta}})$ | ✓ | ✗ | ✗ |
| Praque Luo et al. (2020) | Non-convex | Lip. | Bou. | N/A | ✓ | ✗ | ✗ |
| DSGD-AAU Xiong et al. (2023) | Non-convex | Lip. | Bou. | $\mathcal{O}(\frac{1}{\sqrt{T}})$ | ✓ | ✗ | ✗ |
| DGD-ATC Wu et al. (2023) | Str-con | Lip. | Unbou. | $\mathcal{O}(\rho^T)$ | ✓ | ✗ | ✗ |
| AD-APD Abolfazli et al. (2023) | Convex | Lip. | Bou. | $\mathcal{O}(\frac{1}{T})$ | ✓ | ✗ | ✗ |
| PC-ASGD (This paper) | Weakly Str-con | Lip. | Bou. | $\mathcal{O}(\rho^T + \frac{1}{T} + \frac{1}{\sqrt{T}})$ | ✓ | ✓ | ✓ |
| | Convex | Lip. | Bou. | $\mathcal{O}(\frac{1}{\sqrt{T}} + \frac{1}{T^{1.5}})$ | ✓ | ✓ | ✓ |
| | Non-convex | Lip. | Bou. | $\mathcal{O}(\frac{1}{\sqrt{T}})$ | ✓ | ✓ | ✓ |

Con.Rate: convergence rate, Str-con: strongly convex. Lip.& Bou.: Lipschitz continuous and bounded. Delay Ass.: Delay Assumption. Unbou.: Unbounded. $T$: Total iterations. D.C.: decentralized computation. G.C.: Gradient Compensation. A.S.: Alternant Step, $\rho \in (0,1)$ is a positive constant. Note that the convergence rate of PC-ASGD is obtained by using the constant step size. $\zeta \in (0,1)$.

**Related work**. In the early works on distributed learning with master-slave architecture, Asynchronous Stochastic Gradient Descent (ASGD) algorithm has been proposed Dean et al. (2013), where each local worker continues its training process right after its gradient is added to the global model. The algorithm could tolerate the delay in communication. Later works Agarwal & Duchi (2011); Feyzmahdavian et al. (2015); Recht et al. (2011); Zhuang et al. (2021) extend ASGD to more realistic scenarios and implement the algorithms with a central server and other parallel workers. Typically, since asynchronous algorithms suffer from stale gradients, researchers have proposed algorithms such as DC-ASGD Zheng et al. (2017), adopting

the concept of delay compensation to reduce the impact of staleness and improve the performance of ASGD. For the distributed learning with peer-to-peer architecture, Lian et al. (2017) proposes an algorithm termed AD-PSGD (decentralized ASGD algorithm, aka D-ASGD) that deals with the problem of the stale parameter exchange, as well as presents theoretical analysis for the algorithm performance under bounded delay. Liang et al. (2020) also proposes a similar algorithm with slightly different assumptions. However, these algorithms do not provide empirical or theoretical analysis regarding the impact of delay in detail. Additional works such as using a central agent for control Nair & Gupta (2017), requiring prolonged communication Tsianos & Rabbat (2016), utilizing stochastic primal-dual method Lan et al. (2020), and adopting importance sampling Du et al. (2020), have also been done to address the communication delay in the decentralized setting. More recently, Rigazzi (2019) proposes the DC-s3gd algorithm to enable large-scale decentralized neural network training with the consideration of delay. Zakharov (2020), Venigalla et al. (2020), Chen et al. (2019) and Abbasloo & Chao (2019) also develop algorithms of asynchronous decentralized training for neural networks, while theoretical guarantee is still missing. Asynchronous version of stochastic gradient push (AGP) Assran & Rabbat (2020) is developed to address the asynchronous training in multi-agent framework. The authors claim that AGP is more robust to failing or stalling agents, than the synchronous first-order methods. While the proposed algorithm is only applicable to the strongly convex objectives. To further advance this area, the most recent schemes such as Praque Luo et al. (2020) adopting a partial all-reduce communication primitive, DSGD-AAU Xiong et al. (2023) utilizing an adaptive asynchronous updates, DGD-ATC Wu et al. (2023) extending the Adapt-then-Combine technique from synchronous algorithms, and AD-APD Abolfazli et al. (2023) leveraging accelerated primal-dual algorithm, are developed, but most of them are limited to only (strongly) convex cases. Another line of work based on Federated Learning Dun et al. (2023); Gamboa-Montero et al. (2023); Miao et al. (2023); Xu et al. (2023); Zhang et al. (2023) has also recently received considerable attention, while all proposed approaches essentially rely on a center server, which may threat the privacy of local workers. Different from the aforementioned works, in this study, we specifically present analysis of the impact of the communication delay on convergence error bounds.

**Contributions**. The contributions of this work are specifically as follows:

- *Algorithm Design*. A novel algorithm, called PC-ASGD for distributed learning is proposed to tackle the convergence issues due to the varying communication delays. Built upon ASGD, the PC-ASGD algorithm consists of two steps. While the predicting step leverages the gradient prediction using Taylor expansion to reduce the staleness of the outdated weights, the clipping step selectively drops the outdated weights to alleviate their negative effects. To balance the effects, a tradeoff parameter is introduced to combine these two steps.

- *Convergence guarantee*. We show that with a proper constant step size, PC-ASGD can converge to the *neighborhood* of the optimal solution at a linear rate for weakly strongly-convex functions while at a sublinear rate for both generally convex and nonconvex functions (specific comparisons with other related existing approaches are listed in Table 1). We also model the delay and take it into consideration in the convergence analysis.

- *Verification studies*. PC-ASGD is deployed on distributed GPUs with three datasets CIFAR-10, CIFAR-100, and TinyImageNet by using PreResNet110, DenseNet, ResNet20, and EfficientNet architectures. Our proposed algorithm outperforms the existing delay tolerant algorithms as well as the variants of the proposed algorithm using only the predicting step or the clipping step.

## 2 Formulation and Preliminaries

Consider $N$ agents in a networked system such that their interactions are driven by a graph $\mathcal{G}$, where $\mathcal{G} = \{\mathcal{V}, \mathcal{E}\}$, where $\mathcal{V} = \{1, 2, .., N\}$ indicates the node or agent set, $\mathcal{E} \subseteq \mathcal{V} \times \mathcal{V}$ is the edge set. Throughout the paper, we assume that the graph is undirected and connected. The connection between any two agents $i$ and $j$ can be determined by their physical connections, leading to the communication between them. Traditionally, if agent $j$ is in the neighborhood of agent $i$, they can communicate with each other. Thus, we define the neighborhood for any agent $i$ as $Nb(i) := \{j \in \mathcal{V} | (i, j) \in \mathcal{E} \text{ or } j = i\}$. Rather than considering

synchronization and asynchronization separately, this paper considers both scenarios together by defining the following terminologies.

**Definition 1.** *At a time step $t$, an agent $j$ is called a **reliable neighbor** of the agent $i$ if agent $i$ has the state information of agent $j$ up to $t-1$.*

**Definition 2.** *At a time step $t$, an agent $j$ is called an **unreliable neighbor** of the agent $i$ if agent $i$ has the state information of agent $j$ only up to $t-\tau$, where $\tau$ is the so-called delay and $1 < \tau < \infty$.*

**Remark:** Definitions 1 and 2 allow us to perceive the delay problem in the decentralized learning with a new perspective that depends on the amount of delay. One agent can selectively make use of the outdated information from unreliable neighbors or completely drop such information. The first scenario is related to most previous works on asynchronous delay tolerant approaches as it involves a gradient prediction technique to reduce the negative effects of stale parameters. The second scenario corresponds to most synchronous schemes since the agent only collects information from the reliable neighbors. In this paper, we mainly focus on the fixed delay with a connected graph to better characterize the influence of delay. Real cases with complex topology and time-varying delay modeling are essential but future extensions of the work.

Thus, inside the neighborhood of an agent, there are reliable and unreliable neighbors respectively. This work aims at studying how to effectively tackle issues such as negative impacts that delays may bring on the performance. We define a set for reliable neighbors of agent $i$ as: $\mathcal{R} := \{j \in Nb(i) \mid \boldsymbol{Pr}(x^j = x^j_{t-1}|t) = 1\}$, implying that agent $j$ has the state information $x$ up to the time $t-1$, i.e., $x^j_{t-1}$. We can directly have the set for unreliable neighbors such that $\mathcal{R}^c = Nb \setminus \mathcal{R}$ [1].

Then we can consider the decentralized empirical risk minimization problems, which can be expressed as the summation of all local losses incurred by each agent:

$$\min \ F(\mathbf{x}) := \sum_{i=1}^{N} \sum_{s \in \mathcal{D}_i} f_i^s(x) \tag{1}$$

where $\mathbf{x} = [x^1; x^2; \ldots; x^N]$, $x^i$ is the local copy of $x \in \mathbb{R}^d$, $\mathcal{D}_i$ is a local data set uniquely known by agent $i$, $f_i^s : \mathbb{R}^d \to \mathbb{R}$ is the incurred local loss of agent $i$ given a sample $s$. Based on the above formulation, we then assume everywhere that our objective function is bounded from below and denote the minimum by $F^* := F(\mathbf{x}^*)$ where $\mathbf{x}^* := \operatorname{argmin} F(\mathbf{x})$. Hence $F^* > -\infty$. Moreover, all vector norms refer to the Euclidean norm while matrix norms refer to the Frobenius norm. Some necessary definitions and assumptions are given below for characterizing the main results.

**Assumption 1.** *Each objective function $f_i$ is assumed to satisfy the following conditions: a) $f_i$ is $\gamma_i-smooth$; b) $f_i$ is proper (not everywhere infinite) and coercive.*

**Assumption 2.** *A mixing matrix $\underline{W} \in \mathbb{R}^{N \times N}$ satisfies a) $\mathbf{1}^\top \underline{W} = \mathbf{1}^\top, \underline{W}\mathbf{1}^\top = \mathbf{1}^\top$; b) $Null\{I - \underline{W}\} = Span\{\mathbf{1}\}$; c) $I \succeq \underline{W} \succ -I$.*

**Assumption 3.** *The stochastic gradient of $F$ at any $\mathbf{x}$ is denoted by $\mathbf{g}(\mathbf{x})$, such that a) $\mathbf{g}(\mathbf{x})$ is the unbiased estimate of gradient $\nabla F(\mathbf{x})$; b) The variance is uniformly bounded by $\sigma^2$, i.e., $\mathbb{E}[\|\mathbf{g}(\mathbf{x}) - \nabla F(\mathbf{x})\|^2] \leq \sigma^2$; c) The second moment of $\mathbf{g}(\mathbf{x})$ is bounded, i.e., $\mathbb{E}[\|\mathbf{g}(\mathbf{x})\|^2] \leq G^2$.*

**Remark:** Given Assumption 1, one immediate consequence is that $F$ is $\gamma_m := \max\{\gamma_1, \gamma_2, \ldots, \gamma_N\}$-smooth at all $\mathbf{x} \in \mathbb{R}^{dN}$. The main outcome of Assumption 2 is that the mixing matrix $\underline{W}$ is doubly stochastic matrix and that we have $e_1(\underline{W}) = 1 > e_2(\underline{W}) \geq .. \geq e_N(\underline{W}) > -1$, where $e_z(\underline{W})$ denotes the $z$-th largest eigenvalue of $\underline{W}$ Zeng & Yin (2018). In Assumption 3, the first two are quite generic. While the third part is much weaker than the bounded gradient that is not necessarily applicable to quadratic-like objectives.

---

[1]Note that the delay varies in the asynchronous learning scheme, and there are two types of asynchronization, (i) fixed value of delays Zheng et al. (2017); Rigazzi (2019) and (ii) time-varying delays Dean et al. (2013); Lian et al. (2017) along the learning process. We follow the first setting in this work to implement the experiments.

# 3 PC-ASGD

## 3.1 Algorithm Design

We present the specific update law for our proposed method, PC-ASGD. In Algorithm 1, for the predicting step (line 6), any agent $k$ that is unreliable has a delay when communicating its weights with agent $i$. To compensate for the delay, we adopt the Taylor expansion to approximate the gradient for each time step. The predicted gradient (or delay compensated gradient) is denoted by $g_k^{dc}(x_{t-\tau}^k)$, which is expressed as follows

$$g_k^{dc}(x_{t-\tau}^k) = \sum_{r=0}^{\tau-1} g_k(x_{t-\tau}^k) + \lambda g_k(x_{t-\tau}^k) \odot g_k(x_{t-\tau}^k) \odot (x_{t-\tau+r}^i - x_{t-\tau}^i), \tag{2}$$

where $\lambda$ is a positive constant in $(0,1]$ and the term $\lambda g_k(x_{t-\tau}^k) \odot g_k(x_{t-\tau}^k)$ is an estimate of the Hessian matrix, $\nabla g_k(x_{t-\tau}^k)$. Throughout the rest of analysis, we define $g_k^{dc,r}(x_{t-\tau}^k) := g_k(x_{t-\tau}^k) + \lambda g_k(x_{t-\tau}^k) \odot g_k(x_{t-\tau}^k) \odot (x_{t-\tau+r}^i - x_{t-\tau}^i)$. We briefly provide explanation for the ease of understanding, while referring interested readers to Appendix A.2 for the details of the derivation of Eq. 2. For agent $k$, at the current time step $t$, since it did not get updated over the past $\tau$ time steps, it is known that $x_t^k := x_{t-\tau}^k$. By abuse of notation, we use $g_k^{dc}(x_{t-\tau}^k)$ instead of $g_k^{dc}(x_t^k)$ for the predicted gradient, as the former reasonably justifies Eq. 2. While we still keep the parameter of $x_t^k$ in the predicting step since it will be convenient for us to derive the compact step in the following. However, one can also replace $x_t^k$ with $x_{t-\tau}^k$ if necessary. Additionally, the communication scheme for the predicting step among agents remains similar as in Jiang et al. (2017) but our method needs extra communication overhead in the predicting step for the gradient information of $g_k(x_{t-\tau}^k)$. This burden can be further reduced by gradient quantization Alistarh et al. (2017) but beyond the scope of our work.

**Remark:** Also, the term $(x_{t-\tau+r}^i - x_{t-\tau}^i)$ is from agent $i$ due to the inaccessible outdated information of agent $k$, which intuitively illustrates that the compensation is driven by the agent $i$ when agent $k$ is in its neighborhood and deemed an unreliable one. We remark that the replacement by using agent $i$ is reasonably feasible due to the following three reasons. First, the individual model difference decays along with time for all agents in the network. Particularly, as agent $k$ is in the neighborhood of agent $i$, the decaying trend between $(x_{t-\tau+r}^i - x_{t-\tau}^i)$ and $(x_{t-\tau+r}^k - x_{t-\tau}^k)$ should be similar. Second, such a replacement may cause an extra error term in the error bound if imposing an assumption to bound the difference, e.g., $\|(x_{t-\tau+r}^i - x_{t-\tau}^i) - (x_{t-\tau+r}^k - x_{t-\tau}^k)\| \leq c$, where $c \geq 0$, but the overall convergence rate remains the same as the step size plays a role to determine it. Finally, the empirical results show the feasibility of such a replacement. If the replacement caused the divergence of the gradient norm, the training loss and testing accuracy would be poor. In our practice, we carefully examine their difference to make sure the replacement proceeds well. However, due to the replacement, the predicted gradient is approximate.

On the contrary, at the time instant $t$, when the clipping step is taken, intuitively, we have to clip the agents that possess outdated information, resulting in the change of the mixing matrix $\underline{W}$. Essentially, we can manipulate the corresponding weight values $w_{ij}, j \in \mathcal{R}^c$ in $\underline{W}$ such that at the clipping step, $w_{ij} = 0, j \in \mathcal{R}^c$. For the convenience of analysis, we introduce $\underline{W}$ to represent the mixing matrix at this step. This seems a bit unnecessary in terms of formulation as the first part in the predicting step can be used for the clipping step, while for analysis it would help clarify, particularly when we show the convex combination in line 8. Additionally, though the first part in the predicting step is essentially the same as the clipping step, the separation between them in Algorithm 1 would make the presentation clear about that these two steps take place in a single update.

Different from the DC-ASGD, which significantly relies on a central server to receive information from each agent, our work removes the dependence on the central server, and instead constructs a graph for all of agents. The clipping step (line 7) essentially rejects information from all the unreliable neighbors in the neighborhood of one agent. Subsequently, the equality in line 8 balances the tradeoff between the predicting and clipping steps. In practice, the determination of $\theta_t$ results in some practical variants. In the empirical study presented in Section 5, one can see that $\theta_t$ is either 0 or 1 by leveraging one condition, which implies that in each epoch, only one step is adopted, yielding two other variants shown in the experiments, C-ASGD

---

**Algorithm 1:** PC-ASGD

---

**Input:** number of agents $N$, learning rate $\eta > 0$, agent interaction matrices $\underline{W}$, $\underline{\tilde{W}}$, number of epochs $T$,
   the tradeoff parameter $0 \leq \theta_t \leq 1, t \in \{0, 1, \ldots, T-1\}$
**Output:** the models' parameters in agents $x_T^i, i = 1, 2, \cdots N$

1: **Initialize** all the agents' parameters $x_0^i$, $i = 1, 2, \cdots N$
2: Do broadcast to identify the clusters of reliable agents and the delay $\tau$
3: $t = 0$
4: **while** *epoch* $t < T$ **do**
5:   **for** each agent $i$ **do**
6:     Predicting Step: $x_{t+1,pre}^i = \sum_{j \in \mathcal{R}} w_{ij} x_t^j - \eta g_i(x_t^i) + \sum_{k \in \mathcal{R}^c} w_{ik}(x_t^k - \eta g_k^{dc}(x_{t-\tau}^k))$
7:     Clipping Step: $x_{t+1,cli}^i = \sum_{j \in Nb(i)} \tilde{w}_{ij} x_t^j - \eta g_i(x_t^i)$
8:     $x_{t+1}^i = \theta_t x_{t+1,pre}^i + (1-\theta_t) x_{t+1,cli}^i$
9:   **end for**
10:   $t = t + 1$
11: **end while**

---

or P-ASGD. However, for the sake of generalization, we provide the analysis for the combined steps (line 8). In practice, we try a practical strategy for adaptive $\theta$ choices and we also show the effectiveness empirically.

Since the term $\sum_{k \in \mathcal{R}^c} w_{ik} g_k^{dc}(x_{t-\tau}^k)$ applies to unreliable neighbors only, for the convenience of analysis, we expand it to the whole graph. It means that we establish an expanded graph to cover all of agents by setting some elements in the mixing matrix $\underline{W}' \in \mathbb{R}^{N \times N}$ equal to 0, but keeping the same connections as in $\underline{W}$. Namely, we have $w_{ik}' = 0, k \in \mathcal{R}$ and $w_{ik}' = w_{ik}, k \in \mathcal{R}^c$. By setting the current time as $t + \tau$, the compact form in line 8 can be rewritten as:

$$\mathbf{x}_{t+\tau+1} = \mathcal{W}_{t+\tau} \mathbf{x}_{t+\tau} - \eta(\mathbf{g}(\mathbf{x}_{t+\tau}) + \theta_{t+\tau} \sum_{r=0}^{\tau-1} W' \mathbf{g}^{dc,r}(\mathbf{x}_t)) \tag{3}$$

$\mathcal{W}_{t+\tau}$ is denoted by $\theta_{t+\tau} W + (1 - \theta_{t+\tau})\tilde{W}$, where $W = \underline{W} \otimes I_{d \times d}$, $\tilde{W} = \underline{\tilde{W}} \otimes I_{d \times d}$, and $W' = \underline{W}' \otimes I_{d \times d}$. We have deferred the derivation of Eq. 3 to the Appendix.

## 4 Convergence Analysis

This section presents convergence results for the PC-ASGD. We show the consensus estimate and the optimality for both weakly strongly-convex (Polyak-Łojasiewicz Condition Karimi et al. (2016)), generally convex, and nonconvex smooth objectives. The consensus among agents (aka, disagreement estimate) can be thought of as the norms $\|x_t^i - x_t^j\|$, the differences between the iterates $x_t^i$ and $x_t^j$. Alternatively, the consensus can be measured with respect to a reference sequence, i.e., $y_t = \frac{1}{N} \sum_{i=1}^{N} x_t^i$. In particular, we discuss $\|x_t^i - y_t\|$ for any time $t$ as the metrics with respect to the delay $\tau$.

**Lemma 1.** *(**Consensus**) Let Assumptions 2 and 3 hold. Assume that the delay compensated gradients are uniformly bounded, i.e., there exists a scalar $B > 0$, such that*

$$\|\mathbf{g}^{dc,r}(\mathbf{x}_t)\| \leq B, \quad \forall t \geq 0 \text{ and } 0 \leq r \leq \tau - 1,$$

*Then for all $i \in V$ and $t \geq 0$, $\exists \eta > 0$, we have*

$$\mathbb{E}[\|x_t^i - y_t\|] \leq \eta \frac{G + (\tau - 1)B\theta_m}{1 - \delta_2}, \tag{4}$$

*where $\theta_m = max\{\theta_{s+1}\}_{s=t}^{t+\tau-1}$, $\delta_2 = max\{\theta_s e_2 + (1 - \theta_s)\tilde{e}_2\}_{s=0}^{t+\tau-1} < 1$, where $e_2 := e_2(W) < 1$ and $\tilde{e}_2 := e_2(\tilde{W}) < 1$.*

The detailed proof is shown in the Appendix. Lemma 1 states the consensus bound among agents, which is proportional to the step size $\eta$ and inversely proportional to the gap between the largest and the second-largest magnitude eigenvalues of the equivalent graph $\mathcal{W}$.

**Remark:** One implication that can be made from Lemma 1 is when $\tau = 1$, the consensus bound becomes the smallest, which can be obtained as $\frac{\eta G}{1-\delta_2}$. This bound is the same as obtained already by most decentralized learning (or optimization) algorithms. This accordingly implies that the delay compensated gradient or predicted gradient does not necessarily require many time steps. Otherwise, more compounding error could be included. Alternatively, $\theta_m = 0$ can also result in such a bound, suggesting that the clipping step dominates in the update. On the other hand, once $\tau \gg 1$ and $\theta_m \neq 0$, the consensus bound becomes worse, which will be validated by the empirical results. Additionally, if the network is sparse, which suggests $e_2 \to 1$ and $\tilde{e}_2 \to 1$, the consensus among agents may not be achieved well and correspondingly the optimality would be negatively affected, which has been justified in existing works Jiang et al. (2017).

Most previous works have typically explored the convergence rate on the strongly convex objectives. However, the assumption of strong convexity can be quite strong in most models such that the results obtained may be theoretically instructive and useful. Hence, we introduce a condition that is able to relax the strong convexity, but still maintain the similar theoretical property, i.e., Polyak-Łojasiewicz (PL) condition Karimi et al. (2016). The condition is expressed as follows: A differentiable function $F$ satisfies the PL condition such that there exists a constant $\mu > 0$

$$\frac{1}{2}\|\nabla F(\mathbf{x})\|^2 \geq \mu(F(\mathbf{x}) - F^*). \tag{5}$$

When $F(\mathbf{x})$ is strongly convex, it also implies the PL condition. However, this is not vice versa. We now state the first main result.

**Theorem 1.** *Let Assumptions 1,2 and 3 hold. Assume that the delay compensated gradients are uniformly bounded, i.e., there exists a scalar $B > 0$ such that*

$$\|\mathbf{g}^{dc,r}(\mathbf{x}_t)\| \leq B, \quad \forall t \geq 0 \ and \ 0 \leq r \leq \tau - 1, \tag{6}$$

*and that $\nabla F(\mathbf{x}_t)$ is $\xi_m$-smooth for all $t \geq 0$. Then for the iterates generated by PC-ASGD, when $0 < \eta \leq \frac{1}{2\mu\tau}$ and the objective satisfies the PL condition, they satisfy*

$$\mathbb{E}[F(\mathbf{x}_t) - F^*] \leq (1 - 2\mu\eta\tau)^{t-1}(F(\mathbf{x}_1) - F^* - \frac{Q}{2\mu\eta\tau}) + \frac{Q}{2\mu\eta\tau}, \tag{7}$$

*where*

$$Q = 2(1 - 2\mu\eta\tau)G\eta C_1 + \frac{\eta^3 \xi_m G}{2}\sum_{r=1}^{\tau-1}C_r + 2\eta^2 G\gamma_m C_1$$

$$+ G\eta\tau\sigma + \eta^2 G(\gamma_m + \epsilon_D + \epsilon + (1-\lambda)G^2)\sum_{r=1}^{\tau-1}C_r + \eta G^2 + \eta^2 \gamma_m G\tau C_2 \tag{8}$$

*and $C_1 = \frac{G+(\tau-1)B\theta_m}{1-\delta_2}, C_r = \frac{2G+(r-1)B\theta_m}{1-\delta_2}, C_2 = \frac{2G+(\tau-1)B\theta_m}{1-\delta_2}$. $\epsilon_D > 0$ and $\epsilon > 0$ are upper bounds for the approximation errors of the Hessian matrix that can be obtained as we describe in the Appendix[2].*

**Remark:** One implication from Theorem 1 is that PC-ASGD enables the iterates $\{\mathbf{x}_t\}$ to converge to the neighborhood of $\mathbf{x}^*$, which is $\frac{Q}{2\eta\mu\tau}$, matching the results by Jiang et al. (2017); Bottou et al. (2018); Patrascu & Necoara (2017). In addition, Theorem 1 shows that the error bound is significantly attributed to network errors caused by the disagreement among agents with respect to the delay and the variance of stochastic gradients. Another implication can be made from Theorem 1 is that the convergence rate is closely related to the delay and the step size such that when the delay is large it may reduce the coefficient, $1 - 2\mu\eta\tau$, to speed up the convergence. However, correspondingly the upper bound of the step size is also reduced. Hence, there is a tradeoff between the step size and the delay in PC-ASGD. Theorem 1 also suggests that when the objective function only satisfies the PL condition and is smooth, the convergence to the neighborhood of $\mathbf{x}^*$ in a linear rate can still be achieved. The PL condition may not necessarily imply convexity and hence the conclusion can even apply to some nonconvex functions. To further analyze the error bound, we define

---

[2]The proof for this theorem is fairly non-trivial and technical. We refer readers to the Appendix for more details. To simplify the proof, this main result will be divided into several lemmas.

$\eta = \mathcal{O}(\frac{1}{\sqrt{t}})$, PC-ASGD enjoys a convergence rate of $\mathcal{O}(\rho^t + \frac{1}{t} + \frac{1}{\sqrt{t}})$ to the neighborhood of $\mathbf{x}^*$, which becomes $G(2(1 - 2\mu\eta\tau)C_1 + \tau\sigma + G)$.

Studying the convergence behavior of PC-ASGD for generally convex functions is critically vital as many objectives in machine learning fall into this category of interest. However, the proof techniques for showing the convergence are different from those shown for the above weakly strongly convex objectives. In the sequel, a well-known result regarding convexity is first introduced and then the theoretical claim is presented, while its associated proof is deferred to the Appendix.

**Lemma 2.** *If $F : \mathbb{R}^d \to \mathbb{R}$ is convex and differentiable, then for all $x, y \in \mathbb{R}^d$, the following holds:*

$$F(x) \geq f(y) + \langle \nabla F(y), x - y \rangle. \tag{9}$$

**Theorem 2.** *Let Assumptions 1, 2 and 3 hold. Assume that the delay compensated gradients are uniformly bounded, i.e., there exists a scalar $B > 0$ such that for all $T \geq 1$*

$$\|\mathbf{g}^{dc}(\mathbf{x}_t)\| \leq B, \quad \forall t \geq 0 \ and \ 0 \leq r \leq \tau - 1, \tag{10}$$

*and there exists $C > 0$,*

$$\mathbb{E}[\|\mathbf{x}_t - \mathbf{x}^*\|] \leq C, \tag{11}$$

*where $\mathbf{x}^* \in \mathrm{argmin} F(\mathbf{x})$. Then for the iterations generated by PC-ASGD, there exists $0 < \eta < \frac{1}{20\gamma_m}$, such that*

$$\mathbb{E}[F(\bar{\mathbf{x}}_T) - F^*] \leq \frac{\|\mathbf{x}_1 - \mathbf{x}^*\|^2}{T\eta} + \frac{A}{\eta}, \tag{12}$$

*where $A = 10\eta^2\sigma_*^2 + 10\eta^2\sigma^2 + 20\eta^4 G^2 C_1^2 + 5\eta^2\theta_m^2\tau^2 B^2 + 2\eta C\theta_m\tau B + 2G\eta^2 C_1(2C+1), C_1 = \frac{G+(\tau-1)B\theta_m}{1-\delta_2}, \sigma_*^2 := \mathbb{E}\|\mathbf{g}(\mathbf{x}^*) - \nabla F(\mathbf{x}^*)\|^2, \bar{\mathbf{x}}_T := \frac{1}{T}\sum_{t=1}^T \mathbf{x}_t.$*

**Remark:** As Theorem 2 suggests, when $F$ is generally convex, asymptotically, PC-ASGD yields the convergence of the iterates $\{\mathbf{x}_t\}$ to the neighborhood of $\mathbf{x}^*$, which is $\frac{A}{\eta}$. Analogously, Theorem 2 shows that the error bound is highly correlated with the consensus estimates among agents and the stochastic gradient variances as well as the time delay. To further analyze the error bound, we still define the $\eta = \mathcal{O}(\frac{1}{\sqrt{T}})$, PC-ASGD exhibits a convergence rate of $\mathcal{O}(\frac{1}{\sqrt{T}} + \frac{1}{T^{1.5}})$ to the neighborhood of $\mathbf{x}^*$, which is $2C(\tau - 1)\theta_m B$. This implies that if there is no time delay, PC-ASGD converges to the optimal solution exactly. Also, the convergence rate matches the state-of-the-art in both centralized Garrigos & Gower (2023); Khaled et al. (2023) and distributed Nedic (2020); Sun et al. (2023); Choi & Kim (2023) settings. In Theorem 2, we also impose a constraint for the distance between $\mathbf{x}_t$ and $\mathbf{x}^*$. Immediately, we know that the error bound in Eq. 12 can become $\frac{C^2}{T\eta} + \frac{A}{\eta}$, which is relatively looser. On the other hand, without such a constraint, we can probably just use $\|\mathbf{x}_t - \mathbf{x}^*\| \leq \|\mathbf{x}_1 - \mathbf{x}^*\|$ to replace $C$, which enables a slightly tighter bound. This also illustrates that Eq. 11 is not a strong constraint.

We next investigate the convergence for the non-convex objectives. For PC-ASGD, we show that it converges to a first-order stationary point at a sublinear rate. It should be noted that such a result may not absolutely guarantee a feasible minimizer due to the lack of some necessary second-order information. However, for most nonconvex optimization problems, this is generic, though some existing works have discussed the second-order stationary points Carmon et al. (2018), which is out of our investigation scope.

**Theorem 3.** *Let Assumptions 1, 2 and 3 hold. Assume that the delay compensated gradients are uniformly bounded, i.e., there exists a scalar $B > 0$ such that for all $T \geq 1$*

$$\|\mathbf{g}^{dc,r}(\mathbf{x}_t)\| \leq B, \quad \forall t \geq 0 \ and \ 0 \leq r \leq \tau - 1, \tag{13}$$

*and there exists $M > 0$,*

$$\mathbb{E}[\|\mathbf{g}^{dc}(\mathbf{x}_t)\|^2] \leq M. \tag{14}$$

*Then for the iterations generated by PC-ASGD, there exists $0 < \eta < \frac{1}{\gamma_m}$, such that*

$$\frac{1}{T}\sum_{t=1}^T \mathbb{E}[\|\nabla F(\mathbf{x}_t)\|^2] \leq \frac{2(F(\mathbf{x}_1) - F^*)}{T\eta} + \frac{R}{\eta}, \tag{15}$$

*where,* $R = 2G\eta^2 C_1 + \frac{\tau^2\eta^2\gamma_m M}{2} + \frac{\eta\sigma^2}{2} + \eta\sigma\tau B + 2\eta^2\gamma_m(\tau B + G)C_1, C_1 = \frac{G+(\tau-1)B\theta_m}{1-\delta_2}$.

**Remark:** Theorem 3 states that with a properly chosen constant step size, PC-ASGD is able to converge the iterates $\{\mathbf{x}_T\}$ to the noisy neighborhood of a stationary point $\mathbf{x}^*$ in a rate of $O(\frac{1}{\sqrt{T}})$, whose radius is determined by $\frac{\sigma^2}{2} + \sigma\tau B$, if we define $\eta = \mathcal{O}(\frac{1}{\sqrt{T}})$. Additionally, based on $\frac{\sigma^2}{2} + \sigma\tau B$, we can know that the error bound is mainly caused by the variance of stochastic gradients and the time delay. As the length of the delay can have an impact on the predicting steps used in the delay compensated gradient, a *short* term prediction may help alleviate the negative effect caused by the stale agents. Otherwise, the compounding error in the delay compensated gradient could deteriorate the performance of the algorithm.

## 5 Experiments

### 5.1 Practical Variant

So far, we have analyzed theoretically in detail how the proposed PC-ASGD converges with some mild assumptions. In practical implementation, we need to choose a suitable $\theta_t$ to enable the training fast with clipping steps and allow the unreliable neighbors to be involved in training with predicting steps. In this context, we develop a heuristic practical variant with a criterion for determining the tradeoff parameter value. Intuitively, if the delay messages from the unreliable neighbors do not influence the training negatively, they should be included in the prediction. This can be determined by the comparison with the algorithm without making use of these messages. The criterion is shown as follows:

$$x_i^{t+1} = \begin{cases} x_{t+1,pre}^i & \frac{\langle x_{t+1,pre}^i - x_t^i, g_i(x_t^i)\rangle}{\|x_{t+1,pre}^i - x_t^i\|} \geq \frac{\langle x_{t+1,cli}^i - x_t^i, g_i(x_t^i)\rangle}{\|x_{t+1,cli}^i - x_t^i\|} \\ x_{t+1,cli}^i & o.w. \end{cases} \tag{16}$$

where we choose the *cosine distance* to compare the distances for predicting and clipping steps. The prediction step is selected if it has the larger cosine distance, which implies that the update due to the predicting step yields the larger loss descent. Otherwise, the clipping step should be chosen by only trusting reliable neighbors. Our practical variant with this criterion still converges since we just set $\theta_t$ as 0 or 1 for each iteration and the previous analysis in our paper still holds. To facilitate the understanding of predicting and clipping steps, in the following experiments, we also have two other variants P-ASGD and C-ASGD. While the former corresponds to an "optimistic" scenario to only rely on the predicting step, the latter presents a "pessimistic" scenario by dropping all outdated agents. Both of variants follow the same convergence rates induced by PC-ASGD. The specific algorithm is shown as Algorithm 2.

### 5.2 Distributed Network and Learning Setting

**Models and Data sets**. D-ASGD is adopted as the baseline algorithm. Two deep learning structures, PreResNet110 He et al. (2016b), DenseNet Huang et al. (2017), ResNet20 He et al. (2016a) and EfficientNet Tan & Le (2019) (noted as *model 1*, *model 2*, *model 3* and *model 4*), are employed. The detailed training settings are illustrated in Appendix C. CIFAR-10, CIFAR-100 and TinyImageNet are used in the experiments following the settings in Krizhevsky (2012). The training data is randomly assigned to each agent, and the parameters of the deep learning structure are maintained within each agent and communicated with the predefined delays. The testing set is utilized for each agent to verify the performance, where our metric is the average accuracy among the agents. 6 runs are carried out for each case and the mean and variance are obtained and listed in Table 3.

**Delay setting**. The delay is set as $\tau$ as discussed before, which means the parameters received from the agents outside of the reliable cluster are the ones that were obtained $\tau$ iterations before. $\tau$ is both fixed at 20 to test the performances of different algorithms including our different variants (P-ASGD, C-ASGD, and PC-ASGD) and baseline algorithms in Section 5.3 and 5.5. We also try to exploit its impact in Section 5.4.

**Distributed network setting**. A distributed network (noted as *distributed network 1*) with 8 agents (nodes) in a fully connected graph is first applied with *model 1-4*, and 2 clusters of reliable agents are defined within the graph consisting of 3 agents and 5 agents, respectively. Then two distributed networks (with

---

**Algorithm 2:** PC-ASGD-PV

---

**Input:** number of agents $N$, learning rate $\eta > 0$, agent interaction matrices $\underline{W}$, $\tilde{\underline{W}}$, number of epochs $T$
**Output:** the models' parameters in agents $x_T^i$, $i = 1, 2, \ldots, N$
 1: **Initialize** all the agents' parameters $x_0^i$, $i = 1, 2, \ldots, N$
 2: Do broadcast to identify the clusters of reliable agents and the delay $\tau$
 3: $t = 0$
 4: **while** *epoch $t < T$* **do**
 5:    **for** each agent $i$ **do**
 6:       Predicting Step: $x_{t+1,pre}^i = \sum_{j \in \mathcal{R}} w_{ij} x_t^j - \eta g_i(x_t^i) + \sum_{k \in \mathcal{R}^c} w_{ik}(x_t^k - \eta g_k^{dc}(x_{t-\tau}^k))$
 7:       Clipping Step: $x_{t+1,cli}^i = \sum_{j \in \mathcal{R}} \tilde{w}_{ij} x_t^j - \eta g_i(x_t^i)$
 8:       $\Delta_{pre} = x_{t+1,pre}^i - x_t^i$; $\Delta_{cli} = x_{t+1,cli}^i - x_t^i$
 9:       **if** $\frac{\langle \Delta_{pre}, g_i(x_t^i) \rangle}{\|\Delta_{pre}\|} \geq \frac{\langle \Delta_{cli}, g_i(x_t^i) \rangle}{\|\Delta_{cli}\|}$ **then**
10:         $x_{t+1}^i = x_{t+1,pre}^i$
11:       **else**
12:         $x_{t+1}^i = x_{t+1,cli}^i$
13:       **end if**
14:    **end for**
15:    $t = t + 1$
16: **end while**

---

5-agent and 20-agent, respectively) are used for scalability analysis, noted as *distributed network 2* and *distributed network 3*, individually. For *distributed network 2*, we construct 2 clusters of reliable agents with 3 and 2 agents. In *distributed network 3*, four clusters are formed and 3 clusters consist of 6 agents while each of the rest has 2 agents.

### 5.3 Performance Evaluation

The testing accuracies on the CIFAR-10 and CIFAR-100 data sets with *model 1* and *model 2* in *distributed network 1* are shown in Fig. 1. It shows that the proposed PC-ASGD outperforms the other single variants and it presents an accuracy increment greater than 2.3% (nearly 4% for DenseNet with CIFAR-10) compared to the baseline algorithm. For other variants P-ASGD or C-ASGD, the testing accuracies are also higher than that of the baseline algorithm. Moreover, PC-ASGD shows faster convergence than P-ASGD as the updating rule overcomes the staleness, and achieves better accuracy than the C-ASGD as it includes the messages from the unreliable neighbors. This is consistent with the analysis in this work. We also show the detailed results of both *distributed network 1* and *distributed network 3* in Table 2.

We then compare our proposed algorithm with other delay-tolerant algorithms in *distributed network 1* with *model 1-4*, including the baseline algorithm D-ASGD, DC-s3gd Rigazzi (2019), D-ASGD with IS Du et al. (2020), and Adaptive Braking Venigalla et al. (2020). The *distributed network 1* is applied for the comparisons. From Table 3, the proposed PC-ASGD obtains the best results in all the cases. It should be noted that some of above-listed algorithms are not designed specifically for this kind of peer-to-peer application (e.g., Adaptive Braking) or may not consider the modeling of severe delays in their works (e.g., D-ASGD with IS and DC-s3gd). In this context, they may not perform well in the test cases. The results also demonstrate our proposed framework can be employed by differential types of models, such as simple ResNet20 and complex EfficientNet. We also conduct numerical studies on TinyImageNet and time-series dataset in Appendix D, the results also verify the effectiveness of our method. Before concluding this section, we remark on the difference between PC-ASGD and DC-s3gd as the latter also leverages the predicting step for the gradient estimates. In our PC-ASGD approach, we differentiate between reliable and unreliable agents, allowing reliable agents to proceed with their updates. For unreliable agents, our method incorporates both predicting and clipping steps, establishing choice criteria to enhance overall performance. In contrast, DC-s3gd exclusively employs prediction (delay compensation) for all agents. Regarding the delay compensation aspect, for agent $i$ requiring an update, our approach employs delay compensation with the delayed gradient

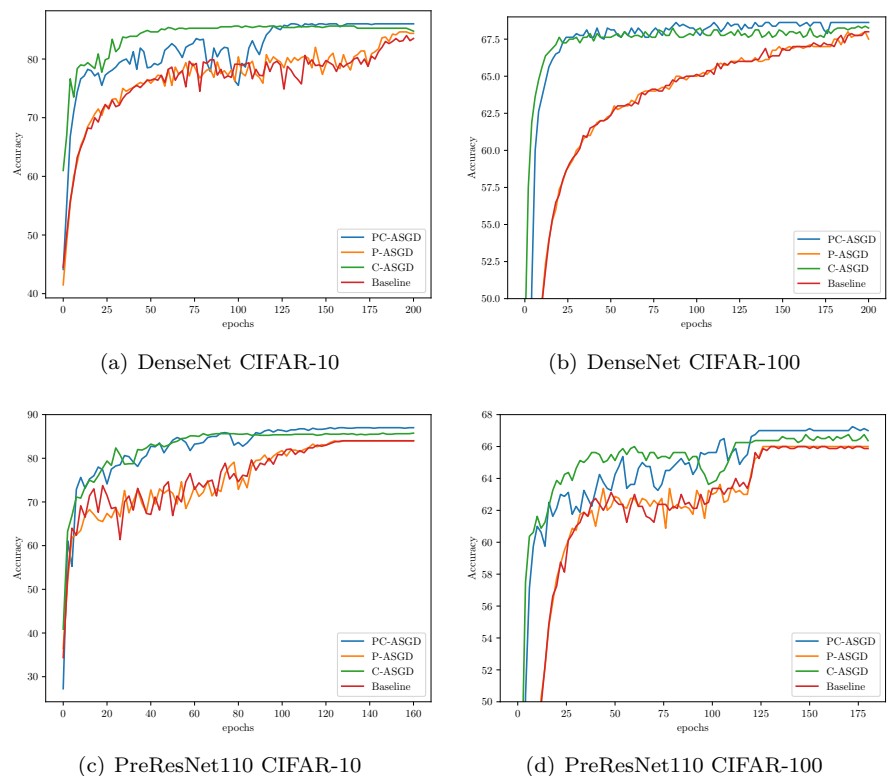

(a) DenseNet CIFAR-10         (b) DenseNet CIFAR-100

(c) PreResNet110 CIFAR-10      (d) PreResNet110 CIFAR-100

Figure 1: Testing accuracy on CIFAR-10 and CIFAR-100 with *distributed network 1.*

Table 2: Performance evaluation of PC-ASGD on CIFAR-10 and CIFAR-100

| | | | | | | | |
|---|---|---|---|---|---|---|---|
| **5 agents** | | | | | | | |
| | PC-ASGD | | P-ASGD | | C-ASGD | | Baseline |
| Model & dataset | acc. (%) | o.p. (%) | acc. (%) | o.p. (%) | acc.(%) | o.p. (%) | acc. (%) |
| Pre110, CIFAR-10 | **87.3 ± 1.1** | **3.3 ± 1.1** | 84.9 ± 0.9 | 0.9 ± 0.9 | 86.0 ± 1.0 | 2.0 ± 1.0 | 84.0 ± 0.3 |
| Pre110, CIFAR-100 | **67.4 ± 1.4** | **3.1 ± 1.9** | 64.8 ± 1.3 | 1.3 ± 1.5 | 66.4 ± 1.2 | 1.9 ± 1.6 | 64.5 ± 1.5 |
| Des, CIFAR-10 | **86.9 ± 0.9** | **3.6 ± 1.8** | 84.4 ± 0.6 | 1.0 ± 1.5 | 85.9 ± 0.9 | 2.7 ± 1.7 | 83.3 ± 0.9 |
| Des, CIFAR-100 | **68.6 ± 0.6** | **2.3 ± 1.7** | 66.8 ± 1.5 | 1.6 ± 1.6 | 66.8 ± 1.6 | 1.8 ± 1.6 | 66.1 ± 1.9 |
| **20 agents** | | | | | | | |
| | PC-ASGD | | P-ASGD | | C-ASGD | | Baseline |
| Model & dataset | acc. (%) | o.p. (%) | acc. (%) | o.p. (%) | acc.(%) | o.p. (%) | acc. (%) |
| Pre110, CIFAR-10 | **84.7 ± 0.9** | **4.2 ± 1.0** | 83.3 ± 0.9 | 2.7 ± 0.9 | 82.5 ± 1.0 | 1.9 ± 1.4 | 80.4 ± 0.7 |
| Pre110, CIFAR-100 | **62.4 ± 0.8** | **3.3 ± 2.0** | 61.7 ± 1.0 | 2.0 ± 1.6 | 61.5 ± 1.0 | 2.5 ± 2.3 | 59.3 ± 1.7 |
| Des, CIFAR-10 | **82.9 ± 0.9** | **2.4 ± 0.9** | 82.0 ± 0.7 | 1.4 ± 1.3 | 81.8 ± 0.6 | 1.8 ± 1.0 | 80.1 ± 0.9 |
| Des, CIFAR-100 | **64.5 ± 0.7** | **3.8 ± 1.7** | 62.5 ± 1.3 | 2.9 ± 2.0 | 62.0 ± 1.5 | 1.3 ± 1.4 | 60.4 ± 1.7 |

acc.–accuracy, o.p.–outperformed comparing to baseline.

of agent $k$ in the unreliable cluster, denoted as $g_k^{dc}$, utilizing the term $(x_t^i - x_{t-\tau}^i)$. Conversely, DC-s3gd utilizes $(x_t^i - x_{t-\tau}^k)$ without any theoretical guarantee. We hypothetically claim that though it is feasible for us to leverage the same predicting step employed in DC-s3gd, while in practice, $(x_t^i - x_{t-\tau}^k)$ may cause larger error bound when models between agents $i$ and $k$ are significantly different. We can also compare the algorithm employing only P-step (P-ASGD), as presented in Table 2 (with 5 agents) with DC-s3gd in Table 3. The results also reveal that in certain scenarios (Pre110/Des, CIFAR100), the performance of P-step is better than that of DC-s3gd, which supports our approximation of $g_k^{dc}$ as well.

Table 3: Performance comparison for different algorithms

| Model & dataset | Pre110 CIFAR-10 | Des CIFAR-10 | ResNet20 CIFAR-10 | Pre110 CIFAR-100 | Des CIFAR-100 | EfficientNet CIFAR-100 |
|---|---|---|---|---|---|---|
| PC-ASGD (Ours) | $\mathbf{87.3 \pm 1.1}$ | $\mathbf{86.9 \pm 0.6}$ | $\mathbf{84.9 \pm 0.6}$ | $\mathbf{67.4 \pm 1.4}$ | $\mathbf{68.6 \pm 0.6}$ | $\mathbf{78.5 \pm 1.3}$ |
| D-ASGD Lian et al. (2017) | $84.0 \pm 0.3$ | $82.5 \pm 0.1$ | $83.3 \pm 0.9$ | $64.5 \pm 1.5$ | $66.1 \pm 1.9$ | $74.7 \pm 0.4$ |
| DC-s3gd Rigazzi (2019) | $86.3 \pm 0.8$ | $85.7 \pm 0.8$ | $83.1 \pm 0.7$ | $63.5 \pm 1.7$ | $66.2 \pm 1.3$ | $76.0 \pm 1.1$ |
| D-ASGD with IS Du et al. (2020) | $85.0 \pm 0.3$ | $84.6 \pm 0.4$ | $83.1 \pm 0.5$ | $64.6 \pm 1.2$ | $66.2 \pm 0.8$ | $75.5 \pm 1.4$ |
| Adaptive Braking Venigalla et al. (2020) | $86.8 \pm 0.9$ | $85.3 \pm 1.0$ | $84.3 \pm 0.4$ | $66.5 \pm 1.2$ | $67.3 \pm 1.1$ | $77.3 \pm 0.8$ |

### 5.4 Impacts of Different Delay Settings

To further show our algorithm's effectiveness, we also implement experiments with different delays. As discussed above, a more severe delay could cause a significant drop in the accuracy. More numerical studies with different steps of delay are presented here. The delays are set as $5, 20, 60$ with our PreResNet110 (*model 1*) of 8 agents (synchronous network without delay is also tested). We use CIFAR-10 in the studies and the topology is *distributed network 1*. The results are shown in Fig. 2.

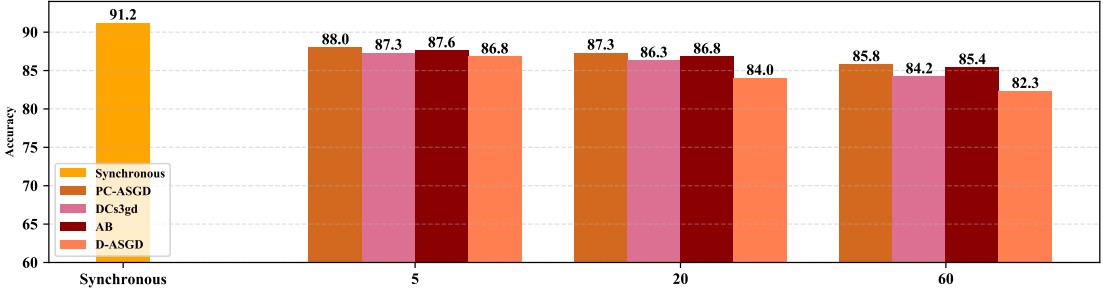

Figure 2: Performance evaluation for different steps of delay.

We can find out as the delay increases, the accuracy decreases. For the synchronous setting, the testing accuracy is close to that in the centralized scenario Yang (2019) but with a higher batch size. When the delay is 60, the accuracy for the D-ASGD reduces significantly, and this validates that the large delay significantly influences the performance and causes difficulties in the training process. However, the delays are practical in real implementations such as industrial IoT platforms. Our proposed PC-ASGD outperforms other algorithms in all cases with different delays. Moreover, the accuracy drop is relatively smaller in cases with larger delays, which suggests that PC-ASGD is more robust to different communication delays.

### 5.5 Impacts of Network Size

For evaluating the performance in different structure sizes of distributed networks, *distributed network 2* and *distributed network 3* follow the same setting as in the *distributed network 1* (delay $\tau = 20$, *model 1*, CIFAR-10). The results are shown in Fig. 3. According to both Table 2 and Fig. 3, as the number of agents increases, the accuracy decreases. It shows that the large size of the network has a negative impact on the training. Our proposed PC-ASGD outperforms all other approaches, which further validates the efficacy and scalability of the proposed algorithm.

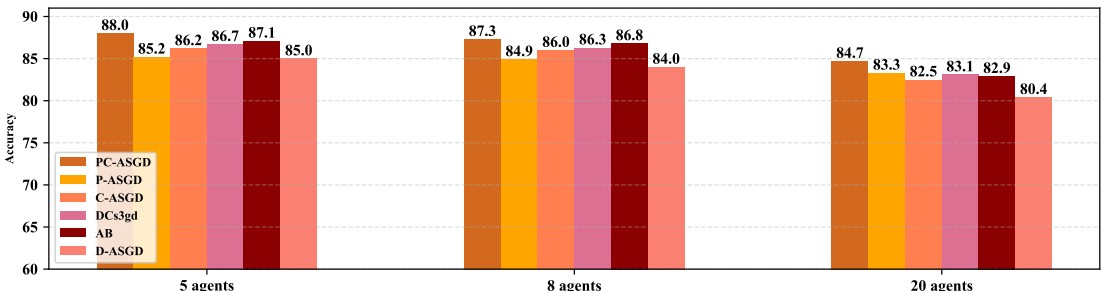

Figure 3: Performance evaluation for different numbers of agents.

## 5.6 Numerical Studies on $\theta$ Assignments

We also conduct empirical studies about the different choices for $\theta$. As we mentioned above, a practical variant is applied for $\theta$, where we intend to form a strategy to determine if the received information (parameters of the deep learning models) is outdated or not. Here, different assignment rules for $\theta$ are tested and compared. *Model 1* is applied, by using CIFAR-10 and the 8 agents system with 3 and 5 agents (*distributed network 1*).

First, $\theta$ is fixed as $0.3, 0.5, 0.7$ (denoted as *f1, f2, f3*), respectively. Then we determine the $\theta$ as $0, 1$ randomly with fixed probability in each round with $0.3, 0.5, 0.7$ (denoted as *p1, p2, p3*). We also try the fully uniformly random assigned $\theta$ in each round (denoted as *r1*). The results are listed in Table 4. The PC-ASGD-PV

Table 4: Mean Performance for Different $\theta$ assignment for Pre110, CIFAR-10

| Method\Parameters | *f1/p1* | *f2/p2* | *f3/p3* |
|---|---|---|---|
| $\theta$ Fixed | 86.3 | 85.0 | 84.5 |
| $\theta$ Bool randomly | 85.6 | 85.0 | 84.1 |
| $\theta$ randomly (r1) | | 85.2 | |
| PC-ASGD-PV | | **87.3** | |
| D-ASGD(Baseline) | | 84.0 | |

obtains the best performance which implies that the trade-off between the predicting step and the clipping step in Algorithm 2 is proper and plays an important role in the convergence process. With the fixed $\theta$ (first row '$\theta$ fixed'), the experimental results show that the optimal ratio between the predicting step and clipping step is 0.3 in this case. And this suggests that more clipping steps are better. For the *p1, p2, p3* cases (second row $\theta$ Bool randomly, i.e. either 0 or 1), the experimental results show that the optimal probability between the predicting step and clipping step is 0.3. This is consistent with the fixed $\theta$ case. Compared with the fix $\theta$ setting, picking $0, 1$ for the $\theta$ in a predefined probability performs worse. The randomness still helps the convergence process but is not as good as the fix $\theta$ setting. For the random $\theta$, the randomness helps the convergence process. However, there exists an optimal $\theta$ for every case and the randomness is not able to get the best performance. The baseline D-ASGD gets the worst performance, which shows the predicting and clipping steps are helpful for the scenarios with delays in the distributed network. This also provides us with the necessity of the additional time cost for the predicting and clipping steps. Note also that optimizing the selection of $\theta$ is beneficial and we can set $\theta$ as binary or non-binary (continuous). The binary setting with the strategy in Algorithm 2 is straightforward and performs well in this work.

To further explore the connection between the $\theta$ selection and the binary strategy in our algorithm, the occurrence of choosing the predicting step or clipping step in PC-ASGD-PV is collected and shown in Fig. 4. The frequencies for the clipping and predicting step choices tend to stabilize with the epochs when the values are around 0.625 and 0.375 respectively. This is consistent with the fixed $\theta$ experiments (where the optimal ratio between the predicting step and clipping step is 0.3, compared to 0.5 and 0.7.) The final choice frequency appears to be empirically determined by the proportion of unreliable clusters. It indicates that the proportion of delayed information included by these unreliable clusters will determine the likelihood of rejecting the P-step. Additionally, we also observe $\theta$ will increase as the time delay diminishes to be smaller

empirically. However, we cannot construct a proportional relationship since we adopt some approximations when dealing with P-step and more theoretical and empirical analysis can be an interesting future direction.

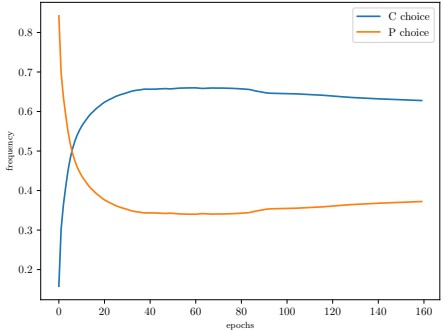

Figure 4: Predicting and clipping steps choices changing with epochs.

## 5.7 Time Cost Comparison

The time cost for the presented algorithm is compared with the baseline algorithm (D-ASGD), P-ASGD, and C-ASGD. The average time costs for *model 1* with CIFAR-10 in *distributed network 1* are collected and shown in Fig. 5. The hardware we adopt is shown in Appendix C.

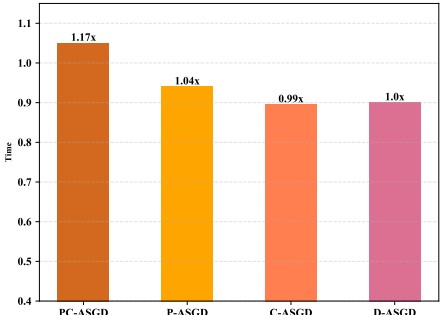

Figure 5: Average time costs for different methods (per epoch).

We observe that the extra time costs for the predicting and clipping steps and additional criterion are not large, although there are still 17% more costs compared to D-ASGD. Therefore, we need to consider the trade-off before implementing the proposed algorithm. However, with the improvement of the local computing resources and the architecture design, the extra time cost might be acceptable with the gains in performance. Moreover, the extra time cost is not changed with the delay, while the boosting in the performance is more significant in large delays (as shown in Fig. 2). It means that our algorithm could be more applicable in the distributed network with various delays, and this is realistic in industrial IoT systems where the computing resources vary remarkably among the agents and the data in each agent also differs significantly.

## 5.8 Validation for Theoretical Analysis

Finally, we present two examples to verify our constructed theoretical analysis. We establish a network involving three agents. We also set two reliable clusters with 1 and 2 agents, respectively. We leverage three nonconvex functions, i.e., Rastrigin, Rosenbrock Liang et al. (2006) and three three-hump camel function Horst et al. (2000) to test the performance of our proposed framework. Though these functions are simple nonconvex problems, they have been used widely to test the performance of many numerical optimizers Mishra (2006). We randomly sample batches during local training in each agent. We set a fixed step size according to our Theorem 2 as 0.008. The number of iterations is set to 500 for each case.

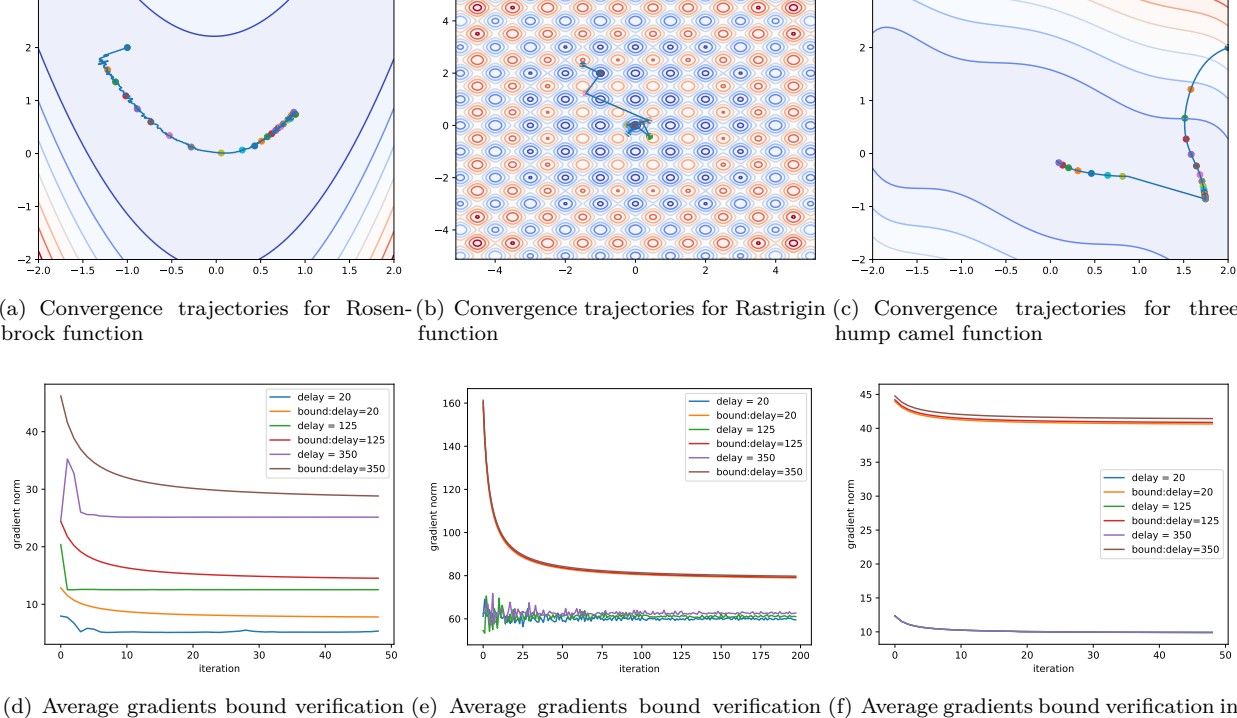

(a) Convergence trajectories for Rosenbrock function

(b) Convergence trajectories for Rastrigin function

(c) Convergence trajectories for three-hump camel function

(d) Average gradients bound verification in Rosenbrock function

(e) Average gradients bound verification in Rastrigin function

(f) Average gradients bound verification in three three-hump camel function

Figure 6: The results of simple functions.

From Fig. 6(a), 6(b) and 6(c), we can view the convergence of our proposed PC-ASGD algorithms. For the bound verification, we take different values of the delay to observe the performances of our theoretical framework. We first find that when the delay is large, the squared norm of the gradient is large, which is consistent with our theoretical analysis. In all three cases, the quantitative results verify the correctness of our proposed theoretical framework in Fig. 6(d), 6(e), and 6(f). In the Rosenbrock function case, our established theory could describe the tendency of the average gradients square norm and the results are nearly tight asymptotically. But in Rastrigin function and three three-hump camel function cases, we observe that the differences between different delays are not large such that the bound is not so tight. However, when calculating bounds, we find that the bounds for different delays differ mildly, which is consistent with all the empirical results. It also shows the effectiveness of our proposed theoretical analysis.

## 6 Limitations

In practical applications, our proposed algorithm still faces challenges. In comparison with the classical D-ASGD, our methods do not excel in communication efficiency. Notably, our methods introduce additional computational overhead, as illustrated in Fig. 5 (approximately 0.17 times). This is attributed to increased communication overhead (doubled for transmitting gradients to compute $g_k^{dc}$) and heightened memory overhead, necessitating agents to store $x_{t-\tau}^i, \dots, x_{t-1}^i$.

These challenges become more pronounced when computational resources, communication bandwidth, and storage capacity are limited. For the real-world implementation of our algorithm, it is necessary to ensure the availability of adequate resources. It's also a promising direction to alleviate these burdens by algorithmic improvements, such as quantization and compression.

In addition to the conventional practice of tuning the learning rate, fine-tuning the parameter $\lambda$ in $g_k^{dc}$ is also essential for optimal results (as detailed in Appendix C). If opting to determine the P/C choice $\theta$, it also requires tuning in accordance with the discussions in Sec. 5.6.

# 7 Conclusion

This paper presents a novel learning algorithm for distributed deep learning with heterogeneous delay characteristics in agent-communication-network systems. We propose PC-ASGD algorithm consisting of a predicting step, a clipping step, and the corresponding update law for reducing the staleness and negative effects caused by the outdated weights. We present theoretical analysis for the convergence rate of the proposed algorithm with constant step size when the objective functions are weakly strongly-convex and nonconvex. The numerical studies show the effectiveness of our proposed algorithms in different distributed systems with delays, by comparing it to multiple baselines. In future work, the cases for distributed networks with diverse delays and dynamic topology will be further studied and tested.

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

# A   Additional Analysis

Before presenting the main results, we introduce some necessary background on the delay compensated gradients.

## A.1   Connection Between PC Steps

As discussed above, PC-ASGD relies upon the two steps to determine the updates for each agent at every time step, as displayed in Fig. 7. We first turn to the clipping step (line 7 of Algorithm 1) where all stale

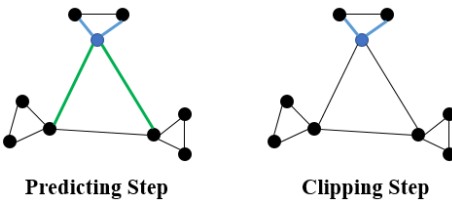

**Predicting Step**          **Clipping Step**

Figure 7: Predicting-Clipping Steps: in the predicting step, blue lines indicate no delay transmission; green lines represent delayed transmission that requires gradient prediction to reduce the stale effect; in the clipping step, the agent selectively drops the delayed information while only receiving information without delay.

information is dropped, which is equivalent to 'clipping' the original graph to become a smaller scale graph. Therefore, between the predicting step and the clipping step, we can observe two static graphs switching alternatively. This also suggests that element values of the mixing matrix $\tilde{W}$ in the clipping step are different from those in the predicting step. In the predicting step (line 6 of Algorithm 1), the agent still requires all the information from its neighbors while asking for gradient prediction from the unreliable neighbors. However, the update is determined by the combination of these two steps in Algorithm 1, which relies on the $\theta$ value to balance the tradeoff. For simplicity, we set the initialization of each agent 0.

We now turn to the practical variant of PC-ASGD in Algorithm 2 in the Appendix. The condition (line 9) adopted for PC-ASGD is based on the approximate cosine value of the angle between $g_i(x_t^i)$ and $\Delta_{pre}$ (or $\Delta_{clip}$). When the angle between $g_i(x_t^i)$ and $\Delta_{pre}$ (or $\Delta_{clip}$) is smaller, leading to a larger cosine value, the corresponding step should be chosen as it enables a larger descent amount along with the direction of $g_i(x_t^i)$. Hence, with a sequence of graphs and the properly set condition, these two alternating steps are connected to each other, allowing for convergence.

## A.2   Delay compensated gradient

We detail how to arrive at Eq. 2. Specifically, given the outdated weights of agent $k$, $x_{t-\tau}^k$, due to the delay equal to $\tau$, by induction, we can obtain for agent $k$

$$
\begin{aligned}
x_{t-\tau+1}^k =& x_{t-\tau}^k - \eta g_k(x_{t-\tau}^k) \\
=& x_{t-\tau}^k - \eta \sum_{r=0}^{0}[g_k(x_{t-\tau}^k) + \lambda g_k(x_{t-\tau}^k) \odot g_k(x_{t-\tau}^k) \odot (x_{t-\tau+r}^i - x_{t-\tau}^i)]
\end{aligned}
\tag{17}
$$

$$
\begin{aligned}
x_{t-\tau+2}^k =& x_{t-\tau+1}^k - \eta g_k(x_{t-\tau+1}^k) = x_{t-\tau}^k - \eta g_k(x_{t-\tau}^k) - \eta g_k(x_{t-\tau+1}^k) \\
\approx& x_{t-\tau}^k - \eta \sum_{r=0}^{1}[g_k(x_{t-\tau}^k) + \lambda g_k(x_{t-\tau}^k) \odot g_k(x_{t-\tau}^k) \odot (x_{t-\tau+r}^i - x_{t-\tau}^i)]
\end{aligned}
\tag{18}
$$

$$\cdots$$

$$
x_t^k \approx x_{t-\tau}^k - \eta \sum_{r=0}^{\tau-1}[g_k(x_{t-\tau}^k) + \lambda g_k(x_{t-\tau}^k) \odot g_k(x_{t-\tau}^k) \odot (x_{t-\tau+r}^i - x_{t-\tau}^i)]
\tag{19}
$$

As we mentioned in the main contents, the term $(x^i_{t-\tau+r} - x^i_{t-\tau})$ is from agent $i$ due to the outdated information of agent $k$, which intuitively illustrates that the compensation is driven by the agent $i$ when agent $k$ is in its neighborhood and deemed an unreliable one.

## A.3   Compact Form of PC Steps

We next briefly discuss how to arrive at the compact form of the predicting and clipping steps for the analysis. For the convenience of analysis, we set the current time step as $t + \tau$ such that line 6 in Algorithm 1 shifts $\tau$ time steps ahead. Let us start with the predicting step and discuss its associated term $\sum_{j \in \mathcal{R}} w_{ij} x^j_{t+\tau} + \sum_{k \in \mathcal{R}^c} w_{ik} x^k_{t+\tau}$, where for the time being, it essentially holds that $x^k_{t+\tau} := x^k_t$. Note that $\mathcal{R}$ includes the agents $i$ itself. Although unreliable neighbors are outdated, in the context, the update for agent $i$ still requires such outdated information, which suggests that the whole graph applies. Additionally, the consensus is performed in parallel with the local computation, so this term boils down to a similar term in the existing consensus-based optimization algorithms in the literature. Thus, one can convert the current consensus term for weights to $\sum_p w_{ip} x^p_{t+\tau}, p \in V$. To show the evolution of predicting gradient over the past steps ranging from 0 to $\tau - 1$, we use $g^{dc,r}_k(x^k_t)$ to represent.

Hence, the update law for the predicting step can be rewritten as:

$$x^i_{t+\tau+1} = \sum_p w_{ip} x^p_{t+\tau} - \eta(g_k(x^i_{t+\tau}) + \sum_{k \in \mathcal{R}^c} w_{ik} \sum_{r=0}^{\tau-1} g^{dc,r}_k(x^k_t)) \tag{20}$$

One may argue that for those outdated agent $k \in \mathcal{R}^c$, they have no information ahead of time $t$, which is $\tau$ time steps back from the current time. As the graph is undirected and connected, the time scale will not change the connections among agents. Also, for agent $i$, it receives always information from other agents, either the current or the outdated to update its weights. Thus, we have,

$$x^p_{t+\tau} = \begin{cases} x^j_{t+\tau} & p = j, j \in \mathcal{R} \\ x^k_t & p = k, k \in \mathcal{R}^c \end{cases} \tag{21}$$

Since the term $\sum_{k \in \mathcal{R}^c} w_{ik} \sum_{r=0}^{\tau-1} g^{dc,r}_k(x^k_t)$ applies to unreliable neighbors only, for the convenience of analysis, we expand it to the whole graph. It means that we establish an expanded graph to cover all of agents by setting some elements in the mixing matrix $\underline{W}' \in \mathbb{R}^{N \times N}$ equal to 0, but keeping the same connections as in $\underline{W}$. Then Eq. 20 can be modified as

$$x^i_{t+\tau+1} = \sum_p w_{ip} x^p_{t+\tau} - \eta(g_k(x^i_{t+\tau}) + \sum_q w'_{iq} \sum_{r=0}^{\tau-1} g^{dc,r}_k(x^q_t)) \tag{22}$$

where

$$w'_{iq} = \begin{cases} w_{ik} & if \ \ q = k, k \in \mathcal{R}^c \\ 0 & if \ \ q \in \mathcal{R} \end{cases} \tag{23}$$

Thus, we know via the above setting that $\underline{W}'$ is at least a row stochastic matrix. We rewrite the update law into a compact form such that

$$\mathbf{x}_{t+\tau+1} = W \mathbf{x}_{t+\tau} - \eta(\mathbf{g}(\mathbf{x}_{t+\tau}) + \sum_{r=0}^{\tau-1} W' \mathbf{g}^{dc,r}(\mathbf{x}_t)). \tag{24}$$

where $W = \underline{W} \otimes I_{d \times d}$ and $W' = \underline{W}' \otimes I_{d \times d}$. Similarly, we rewrite the clipping steps in a vector form as follows:

$$\mathbf{x}_{t+\tau+1} = \tilde{W} \mathbf{x}_{t+\tau} - \eta \mathbf{g}(\mathbf{x}_{t+\tau}) \tag{25}$$

where $\tilde{W} = \underline{\tilde{W}} \otimes I_{d \times d}$. We are now ready to give the generalized step

$$\mathbf{x}_{t+\tau+1} = \mathcal{W}_{t+\tau} \mathbf{x}_{t+\tau} - \eta(\mathbf{g}(\mathbf{x}_{t+\tau}) + \theta_{t+\tau} \sum_{r=0}^{\tau-1} W' \mathbf{g}^{dc,r}(\mathbf{x}_t)), \tag{26}$$

where $\mathcal{W}_{t+\tau}$ is denoted as $\theta_{t+\tau}W + (1 - \theta_{t+\tau})\tilde{W}$ throughout the rest of the analysis. Though the original graphs corresponding to the predicting and clipping steps are static, the equivalent graph $\mathcal{W}_{t+\tau}$ has become time-varying due to the time-varying $\theta$ value.

### A.4 Approximate Hessian Matrix

Based on the update law, we know that the key part of PC-ASGD is the delay compensated gradients using Taylor expansion and Hessian approximation. Therefore, the Taylor expansion of the stochastic gradient $\mathbf{g}(\mathbf{x}_{t+\tau})$ at $\mathbf{x}_t$ can be written as follows:

$$\mathbf{g}(\mathbf{x}_{t+\tau}) = \mathbf{g}(\mathbf{x}_t) + \nabla\mathbf{g}(\mathbf{x}_t)(\mathbf{x}_{t+\tau} - \mathbf{x}_t) + O((\mathbf{x}_{t+\tau} - \mathbf{x}_t)^2)I, \tag{27}$$

where $\nabla\mathbf{g}$ denotes the matrix with the element $\nabla g_{ij} = \frac{\partial F}{\partial x^i \partial x^j}$ for all $i, j \in V$.

In most asynchronous SGD works, they used the zero-order item in Taylor expansion as its approximation to $\mathbf{g}(\mathbf{x}_{t+\tau})$ by ignoring the higher order term. Following from Zheng et al. (2017), we have

$$\mathbf{g}(\mathbf{x}_{t+\tau}) \approx \mathbf{g}(\mathbf{x}_t) + \nabla\mathbf{g}(\mathbf{x}_t)(\mathbf{x}_{t+\tau} - \mathbf{x}_t), \tag{28}$$

Directly adopting the above equation would be difficult in practice since $\nabla\mathbf{g}(\mathbf{x}_t)$ is generically computationally intractable when the model is very large, such as deep neural networks. To make the delay compensated gradients in PC-ASGD technically feasible, we apply approximation techniques for the Hessian matrix. We first use $O(\mathbf{x}_t)$ to denote the outer product matrix of the gradient at $\mathbf{x}_t$, i.e.,

$$O(\mathbf{x}_t) = (\frac{\partial}{\partial\mathbf{x}_t}F(\mathbf{x}_t))(\frac{\partial}{\partial\mathbf{x}_t}F(\mathbf{x}_t))^T \tag{29}$$

When the objective functions take the form of the cross-entropy loss or negative log-likelihood, the outer product of the gradient is an asymptotically unbiased estimation of the Hessian, according to the two equivalent methods to calculate the Fisher information matrix Friedman et al. (2001). That is,

$$\epsilon_t = \mathbb{E}[\|O(\mathbf{x}_t) - H(\mathbf{x}_t)\|] \to 0, \quad t \to 0 \tag{30}$$

where $H(\mathbf{x}_t)$ is the Hessian matrix of $F$ at point $\mathbf{x}_t$.

The above equivalence relies on assumptions that the underlying distribution equals the model distribution with parameter $\mathbf{x}^*$ and that the training model $\mathbf{x}_t$ asymptotically converges to the (globally or locally) optimal model $\mathbf{x}^*$. According to the universal approximation theorem for DNN and some recent results on the optimality of the local optimal, such assumptions are technically reasonable. As the above equivalence was only developed by the negative log-likelihood form, that may not be applicable when we use PC-ASGD for the mean square error form, such as some time-series predictions with LSTM networks. Therefore, we introduce one assumption on the top of such an equivalence as follows,

$$\mathbb{E}[\|O(\mathbf{x}_t) - H(\mathbf{x}_t)\|] \leq \epsilon \quad \exists\epsilon > 0 \tag{31}$$

which primarily eliminates the computational complexity when directly calculating $H(\mathbf{x}_t)$. Another concern would be the large variance probably caused by $O(\mathbf{x}_t)$, though it is an unbiased estimation of $H(\mathbf{x}_t)$. Similar to Zheng et al. (2017), we introduce a new approximator $\lambda O(\mathbf{x}_t) \triangleq \lambda(\frac{\partial}{\partial\mathbf{x}_t}F(\mathbf{x}_t))(\frac{\partial}{\partial\mathbf{x}_t}F(\mathbf{x}_t))^T$. The authors in Zheng et al. (2017) have proved that $\lambda O(\mathbf{x}_t)$ is able to lead to smaller variance during training. Thus we refer interested readers to Zheng et al. (2017) for more details.

To reduce the storage of the approximator $\lambda O(\mathbf{x}_t)$, one widely-used diagonalization trick is adopted Becker & Lecun (1989). Hence, in the update law for PC-ASGD, we can see in the delay compensated gradient involving $\lambda g(\mathbf{x}_t) \odot \lambda g(\mathbf{x}_t)$. By denoting the diagonalized approximator as $Diag(\lambda O(\mathbf{x}_t))$, the following relationship is obtained:

$$Diag(\lambda O(\mathbf{x}_t)) = \lambda g(\mathbf{x}_t) \odot \lambda g(\mathbf{x}_t) \tag{32}$$

However, for analysis, when we apply diagonalization to $H(\mathbf{x}_t)$, it could cause diagonalization error such that we assume that the error is upper bounded by a constant $\epsilon_D > 0$, i.e.,

$$\|Diag(H(\mathbf{x}_t)) - H(\mathbf{x}_t)\| \leq \epsilon_D \tag{33}$$

## B   Additional Proof

For completeness, when presenting proof, we re-present statements for all lemmas and theorems.

**Lemma 3**: The iterates generated by PC-ASGD satisfy $\forall t \geq 0$, and $\tau \geq 2$:

$$\mathbf{x}_{t+\tau} = \prod_{v=0}^{t+\tau-1} \mathcal{W}_{t+\tau-1-v}\mathbf{x}_0 - \eta \sum_{s=0}^{t+\tau-1} \prod_{v=s+1}^{t+\tau-1} \mathcal{W}_{t+\tau+s-v}\mathbf{g}(\mathbf{x}_s) - \eta \sum_{s=t}^{t+\tau-1} \prod_{v=s+1}^{t+\tau-1} \theta_{s+1}\mathcal{W}_{t+\tau+s-v} \sum_{r=0}^{\tau-2} W'\mathbf{g}(\mathbf{x}_{s+1}).$$
$$\tag{34}$$

*Proof.* Based on the vector form of the update law, we obtain

$$\mathbf{x}_{t+\tau} = \mathcal{W}_{t+\tau-1}\mathbf{x}_{t+\tau-1} - \eta(\mathbf{g}(\mathbf{x}_{t+\tau-1}) + \theta_{t+\tau-1} \sum_{r=0}^{\tau-2} W'\mathbf{g}^{dc,r}(\mathbf{x}_t)) \tag{35}$$

With the above equation, it can be observed that $\mathbf{x}_{t+\tau}$ is a function with respect to $\mathbf{x}_t$, which contains all of agents. This suggests that by $\mathbf{x}_t$, there were no delay compensated gradients, while after $\mathbf{x}_{t+1}$, the unreliable neighbors need the delay compensated gradients due to delay. Hence, applying the above equation from 0 to $t + \tau - 1$ yields the desired result. □

**Bounded (stochastic) gradient assumption**: As $\mathbb{E}[\|\mathbf{g}(\mathbf{x})\|^2] \leq G^2$ and $\mathbb{E}[\mathbf{g}(\mathbf{x})] = \nabla F(\mathbf{x})$, one can get that $\|\nabla F(\mathbf{x})\| = \|\mathbb{E}[\mathbf{g}(\mathbf{x})]\| \leq \mathbb{E}[\|\mathbf{g}(\mathbf{x})\|] = \sqrt{(\mathbb{E}[\|\mathbf{g}(\mathbf{x})\|])^2} \leq \sqrt{\mathbb{E}[\|\mathbf{g}(\mathbf{x})\|^2]} = G$.

**Lemma 1**: Let Assumptions 2 and 3 hold. Assume that the delay compensated gradients are uniformly bounded, i.e., there exists a scalar $B > 0$, such that

$$\|\mathbf{g}^{dc,r}(\mathbf{x}_t)\| \leq B, \quad \forall t \geq 0 \ and \ 0 \leq r \leq \tau - 1, \tag{36}$$

Then for all $i \in V$ and $t \geq 0$, $\exists \eta > 0$, we have

$$\mathbb{E}[\|x_t^i - y_t\|] \leq \eta \frac{G + (\tau - 1)B\theta_m}{1 - \delta_2}, \tag{37}$$

where $\theta_m = \max\{\theta_{s+1}\}_{s=t}^{t+\tau-1}$, $\delta_2 = \max\{\theta_s e_2 + (1 - \theta_s)\tilde{e}_2\}_{s=0}^{t+\tau-1} < 1$, where $e_2 := e_2(W) < 1$ and $\tilde{e}_2 := e_2(\tilde{W}) < 1$.

*Proof.* Since

$$\|x_{t+\tau}^i - y_{t+\tau}\| \leq \|\mathbf{x}_{t+\tau} - y_{t+\tau}\mathbf{1}\| = \|\mathbf{x}_{t+\tau} - \frac{1}{N}\mathbf{1}^T\mathbf{x}_{t+\tau}\mathbf{1}\|$$
$$= \|\mathbf{x}_{t+\tau} - \frac{1}{N}\mathbf{1}\mathbf{1}^T\mathbf{x}_{t+\tau}\| = \|(I - \frac{1}{N}\mathbf{1}\mathbf{1}^T)\mathbf{x}_{t+\tau}\|, \tag{38}$$

where $\mathbf{1}$ is the column vector with entries all being 1. According to Assumption 2, we have $\frac{1}{N}\mathbf{1}\mathbf{1}^T\mathcal{W} = \frac{1}{N}\mathbf{1}\mathbf{1}^T$. Hence, by induction, setting $\mathbf{x}_0 = 0$, and Lemma 3, the following relationship can be obtained

$$
\begin{aligned}
&\|\mathbf{x}_{t+\tau} - y_{t+\tau}\mathbf{1}\| \\
=&\eta\|\sum_{s=0}^{t+\tau-1}(\prod_{v=s+1}^{t+\tau-1}\mathcal{W}_{t+\tau+s-v} - \frac{1}{N}\mathbf{1}\mathbf{1}^T)\mathbf{g}(\mathbf{x}_s) + \sum_{s=t}^{t+\tau-1}(\prod_{v=s+1}^{t+\tau-1}\mathcal{W}_{t+\tau+s-v} - \frac{1}{N}\mathbf{1}\mathbf{1}^T)\theta_{s+1}\sum_{r=0}^{\tau-2}W'\mathbf{g}^{dc,r}(\mathbf{x}_t)\| \\
\leq&\eta\sum_{s=0}^{t+\tau-1}\|\prod_{v=s+1}^{t+\tau-1}\mathcal{W}_{t+\tau+s-v} - \frac{1}{N}\mathbf{1}\mathbf{1}^T\|\|\mathbf{g}(\mathbf{x}_s)\| + \eta\sum_{s=t}^{t+\tau-1}\|\prod_{v=s+1}^{t+\tau-1}\mathcal{W}_{t+\tau+s-v} - \frac{1}{N}\mathbf{1}\mathbf{1}^T\|\|\theta_{s+1}\sum_{r=0}^{\tau-2}W'\mathbf{g}^{dc,r}(\mathbf{x}_t)\| \\
\leq&\eta G\sum_{s=0}^{t+\tau-1}\delta_2^{t+\tau-1-s} + \eta\sum_{s=t}^{t+\tau-1}\delta_2^{t+\tau-1-s}\theta_{s+1}(\tau-1)B \\
\leq&\eta G\frac{1}{1-\delta_2} + \eta(\tau-1)B\theta_m\frac{\delta_2^t - \delta_2^{t+\tau-1}}{1-\delta_2} \\
\leq&\eta\frac{G + (\tau-1)B\theta_m}{1-\delta_2}.
\end{aligned}
$$

$$(39)$$

The second inequality follows from the Triangle inequality and Cauthy-Schwartz inequality and the third inequality follows from Assumption 2 and that the matrix $\frac{1}{N}\mathbf{1}\mathbf{1}^T$ is the projection of $\mathcal{W}$ onto the eigenspace associated with the eigenvalue equal to 1. The last inequality follows from the property of geometric sequence. The proof is completed by replacing $t + \tau$ with $t$ on the left hand side. $\qquad\square$

To prove the main results, we present several auxiliary lemmas first. We define

$$
\begin{aligned}
\mathcal{G}^h(\mathbf{x}_t) &= \sum_{r=0}^{\tau-1}\mathbf{g}(\mathbf{x}_{t+r}) + H(\mathbf{x}_t)(\mathbf{v}_{t+r} - \mathbf{x}_t) \\
\nabla\mathcal{F}^h(\mathbf{x}_t) &= \sum_{r=0}^{\tau-1}\nabla F(\mathbf{x}_{t+r}) + \mathbb{E}[H(\mathbf{x}_t)(\mathbf{v}_{t+r} - \mathbf{x}_t)]
\end{aligned}
$$

$$(40)$$

which are the incrementally delay compensated gradient and its expectation, respectively. It can be observed that $\mathcal{G}^h(\mathbf{x}_t)$ is the unbiased estimator of $\nabla\mathcal{F}^h(\mathbf{x}_t)$. It should be noted that $H(\mathbf{x}_t) = \nabla\mathbf{g}(\mathbf{x}_t)$. Let $\mathbf{v}_{t+\tau} = \mathcal{W}_{t+\tau}\mathbf{x}_{t+\tau}$. We next present a lemma to upper bound $\|\nabla F(\mathbf{v}_{t+r}) - \nabla\mathcal{F}^{h,r}(\mathbf{x}_t)\|$, where $\nabla\mathcal{F}^{h,r}(\mathbf{x}_t) = \nabla F(\mathbf{x}_{t+r}) + \mathbb{E}[H(\mathbf{x}_t)(\mathbf{v}_{t+r} - \mathbf{x}_t)]$.

**Lemma 4**: Let Assumptions 1,2 and 3 hold. Assume that $\nabla F(\mathbf{x}_t)$ is $\xi_m$-smooth. For $t \geq 0$, the iterates generated by PC-ASGD satisfy the following relationship, when $r \geq 1$

$$\|\nabla F(\mathbf{v}_{t+r}) - \nabla\mathcal{F}^{h,r}(\mathbf{x}_t)\| \leq \frac{\xi_m}{2}\eta^2\left(\frac{2G + (r-1)B\theta_m}{1-\delta_2}\right)^2;$$

$$(41)$$

when $r = 0$, we have

$$\|\nabla F(\mathbf{v}_t) - \nabla F(\mathbf{x}_t)\| \leq 2\gamma_m\frac{\eta(G + (\tau-1)B\theta_m)}{1-\delta_2}.$$

$$(42)$$

*Proof.* By the smoothness condition for $\nabla F(\mathbf{x})$, we have

$$\|\nabla F(\mathbf{v}_{t+r}) - \nabla\mathcal{F}^{h,r}(\mathbf{x}_t)\| \leq \frac{\xi_m}{2}\|\mathbf{v}_{t+r} - \mathbf{x}_t\|^2 \leq \frac{\xi_m}{2}\|\mathbf{x}_{t+r} - \mathbf{x}_t\|^2$$

$$(43)$$

Let $\boldsymbol{\Delta}_{t+r} = \mathbf{x}_{t+r} - \mathbf{x}_t$. Thus, based on Lemma 1, we have

$$\mathbf{x}_{t+r} = \prod_{v=t}^{t+r-1}\mathcal{W}_{t+r-1-v}\mathbf{x}_t - \eta\sum_{s=t}^{t+r-1}\prod_{v=s+1}^{t+r-1}\mathcal{W}_{t+r+s-v}\mathbf{g}(\mathbf{x}_s) - \eta\sum_{s=t}^{t+r-1}\prod_{v=s+1}^{t+r-1}\mathcal{W}_{t+s+r-v}\sum_{z=0}^{r-2}\theta_{s+1}W'\mathbf{g}^{dc,z}(\mathbf{x}_{s+1-r})$$

$$(44)$$

Hence, we can obtain

$$\|\mathbf{\Delta}_{t+r}\|^2 = \|(\prod_{v=t}^{t+r-1}\mathcal{W}_{t+r-1-v}-I)\mathbf{x}_t - \eta\sum_{s=t}^{t+r-1}\prod_{v=s+1}^{t+r-1}\mathcal{W}_{t+r+s-v}\mathbf{g}(\mathbf{x}_s) - \eta\sum_{s=t}^{t+r-1}\prod_{v=s+1}^{t+r-1}\mathcal{W}_{t+s+r-v}\sum_{z=0}^{r-2}\theta_{s+1}W'\mathbf{g}^{dc,z}(\mathbf{x}_{s+1-r})\|^2$$

(45)

Due to $\mathbf{x}_0 = 0$ and no delay compensated gradients before time step $t$, we can obtain

$$
\begin{aligned}
&\|\mathbf{\Delta}_{t+r}\|^2\\
=&\| - \eta\sum_{s=0}^{t+r-1}\prod_{v=s+1}^{t+r-1}\mathcal{W}_{t+r+s-v}\mathbf{g}(\mathbf{x}_s) - \eta\sum_{s=t}^{t+r-1}\prod_{v=s+1}^{t+r-1}\mathcal{W}_{t+s+r-v}\sum_{z=0}^{r-2}\theta_{s+1}W'\mathbf{g}^{dc,z}(\mathbf{x}_{s+1-r}) + \eta\sum_{s=0}^{t}\prod_{v=s}^{t}\mathcal{W}_{t+s-v}\mathbf{g}(\mathbf{x}_s)\|^2\\
\leq&\eta^2(\|\sum_{s=0}^{t+r-1}\prod_{v=s+1}^{t+r-1}\mathcal{W}_{t+r+s-v}\mathbf{g}(\mathbf{x}_s)\| + \|\sum_{s=t}^{t+r-1}\prod_{v=s+1}^{t+r-1}\mathcal{W}_{t+s+r-v}\sum_{z=0}^{r-2}\theta_{s+1}W'\mathbf{g}^{dc,z}(\mathbf{x}_{s+1-r})\| + \|\sum_{s=0}^{t}\prod_{v=s}^{t}\mathcal{W}_{t+s-v}\mathbf{g}(\mathbf{x}_s)\|)^2\\
\leq&\eta^2(\sum_{s=0}^{t+r-1}\|\prod_{v=s+1}^{t+r-1}\mathcal{W}_{t+r+s-v}\mathbf{g}(\mathbf{x}_s)\| + \sum_{s=t}^{t+r-1}\|\prod_{v=s+1}^{t+r-1}\mathcal{W}_{t+s+r-v}\sum_{z=0}^{r-2}\theta_{s+1}W'\mathbf{g}^{dc,z}(\mathbf{x}_{s+1-r})\| + \sum_{s=0}^{t}\|\prod_{v=s}^{t}\mathcal{W}_{t+s-v}\mathbf{g}(\mathbf{x}_s)\|)^2\\
\leq&\eta^2(\sum_{s=0}^{t+r-1}\prod_{v=s+1}^{t+r-1}\|\mathcal{W}_{t+r+s-v}\|\|\mathbf{g}(\mathbf{x}_s)\| + \sum_{s=t}^{t+r-1}\prod_{v=s+1}^{t+r-1}\|\mathcal{W}_{t+s+r-v}\|\|\sum_{z=0}^{r-2}\theta_{s+1}W'\mathbf{g}^{dc,z}(\mathbf{x}_{s+1-r})\|\\
&+ \sum_{s=0}^{t}\prod_{v=s}^{t}\|\mathcal{W}_{t+s-v}\|\|\mathbf{g}(\mathbf{x}_s)\|)^2\\
\leq&\eta^2(\frac{2G}{1-\delta_2} + \frac{1}{1-\delta_2}B(r-1)\theta_m)^2\\
\leq&\eta^2(\frac{2G+\theta_m(r-1)B}{1-\delta_2})^2
\end{aligned}
$$

(46)

The first inequality follows from the Triangle inequality. The second inequality follows from the Jensen inequality. The third inequality follows from the Cauthy-Schwartz inequality and the submultiplicative matrix norm applied to stochastic matrices. The fourth inequality follows from the Assumption 2 and bounded gradient. We have observed that this holds when $r \geq 1$. While $r = 0$ enables $\|\nabla F(\mathbf{v}_{t+r}) - \mathcal{F}^{h,r}(\mathbf{x}_t)\|$ to degenerate to $\|\nabla F(\mathbf{v}_t) - \nabla F(\mathbf{x}_t)\|$ based on the definition of $\mathcal{F}^h(\mathbf{x}_t)$. Using the smoothness condition of $F(\mathbf{x})$, we can immediately obtain

$$\|\nabla F(\mathbf{v}_t) - \nabla F(\mathbf{x}_t)\| \leq 2\gamma_m\eta\frac{G+(\tau-1)B\theta_m}{1-\delta_2}.$$

(47)

The proof is completed. □

**Lemma 5**: Let Assumptions 1, 2 and 3 hold. Assume that the delay compensated gradients are uniformly bounded, i.e., there exists a scalar $B > 0$ such that

$$\|\mathbf{g}^{dc,r}(\mathbf{x}_t)\| \leq B, \quad \forall t \geq 0 \ and \ 0 \leq r \leq \tau - 1,$$

(48)

Then for the iterates generated by PC-ASGD, $\exists\eta > 0$, they satisfy

$$
\begin{aligned}
&\|\mathbb{E}[\mathcal{G}^h(\mathbf{x}_t)] - \sum_{r=0}^{\tau-1}W'\mathbf{g}^{dc,r}(\mathbf{x}_t)\|\\
&\leq \sum_{r=1}^{\tau-1}(\gamma_m + \epsilon_D + \epsilon + (1-\lambda)G^2)\eta\frac{2G+(r-1)B\theta_m}{1-\delta_2} + \tau\sigma
\end{aligned}
$$

(49)

*Proof.* Based on the definition of $\mathbb{E}\mathcal{G}^h(\mathbf{x}_t)$, we have

$$\|\mathbb{E}[\mathcal{G}^h(\mathbf{x}_t)] - \sum_{r=0}^{\tau-1} W'\mathbf{g}^{dc,r}(\mathbf{x}_t)\| = \|\mathbb{E}[\sum_{r=0}^{\tau-1} \mathbf{g}(\mathbf{x}_{t+r}) + \sum_{r=0}^{\tau-1} H(\mathbf{x}_t)(\mathbf{x}_{t+r} - \mathbf{x}_t)] - \sum_{r=0}^{\tau-1} W'\mathbf{g}^{dc,r}(\mathbf{x}_t)\|$$

$$= \|\mathbb{E}[\mathcal{G}^{h,r=0}(\mathbf{x}_t)] - W'\mathbf{g}^{dc,r=0}(\mathbf{x}_t) + \mathbb{E}[\mathcal{G}^{h,r=1}(\mathbf{x}_t)] - W'\mathbf{g}^{dc,r=1}(\mathbf{x}_t) + \cdots + \mathbb{E}[\mathcal{G}^{h,r=\tau-1}(\mathbf{x}_t)] - W'\mathbf{g}^{dc,r=\tau-1}(\mathbf{x}_t)\|$$

$$\leq \|\mathbb{E}[\mathcal{G}^{h,r=0}(\mathbf{x}_t)] - W'\mathbf{g}^{dc,r=0}(\mathbf{x}_t)\| + \|\mathbb{E}[\mathcal{G}^{h,r=1}(\mathbf{x}_t)] - W'\mathbf{g}^{dc,r=1}(\mathbf{x}_t)\| + \cdots + \|\mathbb{E}[\mathcal{G}^{h,r=\tau-1}(\mathbf{x}_t)] - W'\mathbf{g}^{dc,r=\tau-1}(\mathbf{x}_t)\|$$

$$\tag{50}$$

The last inequality follows from the Triangle inequality. Now let us discuss $\|\mathbb{E}\mathcal{G}^{h,r}(\mathbf{x}_t) - W'\mathbf{g}^{dc,r}(\mathbf{x}_t)\|$. The following analysis is for cases where $r \geq 1$. We give a brief analysis for the case in which $r = 0$ subsequently.

$$\|\mathbb{E}[\mathcal{G}^h(\mathbf{x}_t)] - W'\mathbf{g}^{dc,r}(\mathbf{x}_t)\|$$

$$= \|\mathbb{E}[\mathbf{g}(\mathbf{x}_{t+r}) + H(\mathbf{x}_t)(\mathbf{x}_{t+r} - \mathbf{x}_t)]W'[\mathbf{g}(\mathbf{x}_t) + \lambda\mathbf{g}(\mathbf{x}_t) \odot \mathbf{g}(\mathbf{x}_t) \odot (\mathbf{x}_{t+r} - \mathbf{x}_t)]\|$$

$$= \|\nabla F(\mathbf{x}_{t+r}) - W'\mathbf{g}(\mathbf{x}_t) + [H(\mathbf{x}_t) - \lambda W'\mathbf{g}(\mathbf{x}_t) \odot \mathbf{g}(\mathbf{x}_t)](\mathbf{x}_{t+r} - \mathbf{x}_t)\|$$

$$\leq \|\nabla F(\mathbf{x}_{t+r}) - W'\mathbf{g}(\mathbf{x}_t)\| + \|[H(\mathbf{x}_t) - \lambda W'\mathbf{g}(\mathbf{x}_t) \odot \mathbf{g}(\mathbf{x}_t)](\mathbf{x}_{t+r} - \mathbf{x}_t)\|$$

$$\leq \|\nabla F(\mathbf{x}_{t+r}) - W'\mathbf{g}(\mathbf{x}_t)\| + \|[H(\mathbf{x}_t) - \lambda W'\mathbf{g}(\mathbf{x}_t) \odot \mathbf{g}(\mathbf{x}_t) + \mathbf{g}(\mathbf{x}_t) \odot \mathbf{g}(\mathbf{x}_t) - \mathbf{g}(\mathbf{x}_t) \odot \mathbf{g}(\mathbf{x}_t)$$
$$- Diag(H(\mathbf{x}_t)) + Diag(H(\mathbf{x}_t))](\mathbf{x}_{t+r} - \mathbf{x}_t)\|$$

$$\leq \|\nabla F(\mathbf{x}_{t+r}) - W'\mathbf{g}(\mathbf{x}_t)\| + \|\mathbf{x}_{t+r} - \mathbf{x}_t\|\|(\lambda W'\mathbf{g}(\mathbf{x}_t) \odot \mathbf{g}(\mathbf{x}_t) - \mathbf{g}(\mathbf{x}_t) \odot \mathbf{g}(\mathbf{x}_t)) + (\mathbf{g}(\mathbf{x}_t) \odot \mathbf{g}(\mathbf{x}_t)$$
$$- Diag(H(\mathbf{x}_t))) + (Diag(H(\mathbf{x}_t)) - H(\mathbf{x}_t))\|$$

$$\leq \|\nabla F(\mathbf{x}_{t+r}) - W'\mathbf{g}(\mathbf{x}_t)\| + \|\mathbf{x}_{t+r} - \mathbf{x}_t\|(\|\lambda W'\mathbf{g}(\mathbf{x}_t) \odot \mathbf{g}(\mathbf{x}_t) - \mathbf{g}(\mathbf{x}_t) \odot \mathbf{g}(\mathbf{x}_t)\| + \|\mathbf{g}(\mathbf{x}_t) \odot \mathbf{g}(\mathbf{x}_t)$$
$$- Diag(H(\mathbf{x}_t))\| + \|Diag(H(\mathbf{x}_t)) - H(\mathbf{x}_t)\|)$$

The third inequality follows from Cauthy-Schwarz inequality while the last one follows from the Triangle inequality. It should be noted that when we combine $H(\mathbf{x}_t)(\mathbf{x}_{t+r} - \mathbf{x}_t)$ and $\lambda W'\mathbf{g}(\mathbf{x}_t) \odot \mathbf{g}(\mathbf{x}_t) \odot (\mathbf{x}_{t+r} - \mathbf{x}_t)$, we follow the update law. Since in a rigorously mathematical sense, $\mathbf{g}(\mathbf{x}_t) \odot \mathbf{g}(\mathbf{x}_t)$ should be $\mathbf{g}(\mathbf{x}_t)\mathbf{g}(\mathbf{x}_t)^T$. However, for reducing the computational complexity when implementing the algorithm, as discussed above, we have made the approximation and diagonalization trick. Hence, we assume that $H(\mathbf{x}_t) - \lambda W'\mathbf{g}(\mathbf{x}_t) \odot \mathbf{g}(\mathbf{x}_t)$ can hold for simplicity and convenience.

Then we discuss $\mathbb{E}[\|\nabla F(\mathbf{x}_{t+r}) - W'\mathbf{g}(\mathbf{x}_t)\|]$.

$$\mathbb{E}[\|\nabla F(\mathbf{x}_{t+r}) - W'\mathbf{g}(\mathbf{x}_t)\|] \leq \mathbb{E}[\|\nabla F(\mathbf{x}_{t+r}) - \mathbf{g}(\mathbf{x}_t)\|]$$

$$= \mathbb{E}[\|\nabla F(\mathbf{x}_{t+r}) - \nabla F(\mathbf{x}_t) + \nabla F(\mathbf{x}_t) - \mathbf{g}(\mathbf{x}_t)\|]$$

$$\leq \mathbb{E}[\|\nabla F(\mathbf{x}_{t+r}) - \nabla F(\mathbf{x}_t)\|] + \mathbb{E}[\|\nabla F(\mathbf{x}_t) - \mathbf{g}(\mathbf{x}_t)\|]$$

$$\leq \gamma_m \|\mathbf{x}_{t+r} - \mathbf{x}_t\| + \sqrt{(\mathbb{E}[\|\nabla F(\mathbf{x}_t) - \mathbf{g}(\mathbf{x}_t)\|])^2} \tag{51}$$

$$\leq \gamma_m \eta \frac{2G + (r-1)B\theta_m}{1 - \delta_2} + \sqrt{\mathbb{E}[\|\nabla F(\mathbf{x}_t) - \mathbf{g}(\mathbf{x}_t)\|]^2}$$

$$\leq \gamma_m \eta \frac{2G + (r-1)B\theta_m}{1 - \delta_2} + \sigma$$

Hence, we have

$$\|\mathbb{E}[\mathcal{G}^h(\mathbf{x}_t)] - \sum_{r=0}^{\tau-1} W'\mathbf{g}^{dc,r}(\mathbf{x}_t)\| \leq \gamma_m \eta \frac{2G + (r-1)B\theta_m}{1 - \delta_2} + [(1-\lambda)G^2 + \epsilon_D + \epsilon]\eta \frac{2G + (r-1)B\theta_m}{1 - \delta_2} + \sigma$$

$$= (\gamma_m + \epsilon_D + \epsilon + (1-\lambda)G^2)\eta \frac{2G + (r-1)B\theta_m}{1 - \delta_2} + \sigma$$

$$\tag{52}$$

The above relationship is obtained for cases where $r \geq 1$. There still is $r = 0$ left. For $r = 0$,

$$\|\nabla F(\mathbf{x}_t) - W'\mathbf{g}(\mathbf{x}_t)\| \leq \sigma \tag{53}$$

Thus, combining each upper bound for $\|\mathbb{E}[\mathcal{G}^{h,r}(\mathbf{x}_t)] - W'\mathbf{g}^{dc,r}(\mathbf{x}_t)\|$, we can obtain

$$\|\mathbb{E}[\mathcal{G}^h(\mathbf{x}_t)] - \sum_{r=0}^{\tau-1} W'\mathbf{g}^{dc,r}(\mathbf{x}_t)\| \leq \sum_{r=1}^{\tau-1}(\gamma_m + \epsilon_D + \epsilon + (1-\lambda)G^2)\eta \frac{2G + (r-1)B\theta_m}{1-\delta_2} + \tau\sigma, \tag{54}$$

which completes the proof. $\qquad \square$

**Lemma 6**: Let Assumptions 1, 2 and 3 hold. Assume that the delay compensated gradients are uniformly bounded, i.e., there exists a scalar $B > 0$ such that

$$\|\mathbf{g}^{dc,r}(\mathbf{x}_t)\| \leq B, \quad \forall t \geq 0 \ and \ 0 \leq r \leq \tau - 1, \tag{55}$$

Then for the iterates generated by PC-ASGD, $\exists \eta > 0$, they satisfy

$$F(\mathbf{x}_{t+\tau}) \geq F(\mathbf{v}_{t+\tau}) - 2G\eta \frac{G + (\tau-1)B\theta_m}{1-\delta_2} \tag{56}$$

*Proof.* Due to the convexity, we have

$$\begin{aligned}
F(\mathbf{x}_{t+\tau}) &\geq F(\mathbf{v}_{t+\tau}) + \nabla F(\mathbf{v}_{t+\tau})(\mathbf{x}_{t+\tau} - \mathbf{v}_{t+\tau}) \\
&\geq F(\mathbf{v}_{t+\tau}) - \|\nabla F(\mathbf{v}_{t+\tau})\|\|\mathbf{v}_{t+\tau} - \mathbf{x}_{t+\tau}\| \\
&\geq F(\mathbf{v}_{t+\tau}) - G\|\mathbf{v}_{t+\tau} - \mathbf{x}_{t+\tau}\| \\
&\geq F(\mathbf{v}_{t+\tau}) - G\|\mathbf{v}_{t+\tau} - y_{t+\tau}\mathbf{1} + y_{t+\tau}\mathbf{1} - \mathbf{x}_{t+\tau}\| \\
&\geq F(\mathbf{v}_{t+\tau}) - G(\|\mathbf{v}_{t+\tau} - y_{t+\tau}\mathbf{1}\| + \|y_{t+\tau}\mathbf{1} - \mathbf{x}_{t+\tau}\|) \\
&\geq F(\mathbf{v}_{t+\tau}) - 2G\eta \frac{G + (\tau-1)B\theta_m}{1-\delta_2}
\end{aligned} \tag{57}$$

The second inequality follows from the Cauthy-Schwarz inequality. The proof is completed. $\qquad \square$

**Theorem 1**: Let Assumptions 1,2 and 3 hold. Assume that the delay compensated gradients are uniformly bounded, i.e., there exists a scalar $B > 0$ such that

$$\|\mathbf{g}^{dc,r}(\mathbf{x}_t)\| \leq B, \quad \forall t \geq 0 \ and \ 0 \leq r \leq \tau - 1, \tag{58}$$

and that $\nabla F(\mathbf{x}_t)$ is $\xi_m$-smooth for all $t \geq 0$. Then for the iterates generated by PC-ASGD, when $0 < \eta \leq \frac{1}{2\mu\tau}$ and the objective satisfies the PL condition, they satisfy

$$\mathbb{E}[F(\mathbf{x}_t) - F^*] \leq (1 - 2\mu\eta\tau)^{t-1}(F(\mathbf{x}_1) - F^* - \frac{Q}{2\mu\eta\tau}) + \frac{Q}{2\mu\eta\tau}, \tag{59}$$

$$\begin{aligned}
Q &= 2(1 - 2\mu\eta\tau)G\eta C_1 + \frac{\eta^3\xi_m G}{2}\sum_{r=1}^{\tau-1} C_r + 2\eta^2 G\gamma_m C_1 \\
&+ G\eta\tau\sigma + \eta^2 G(\gamma_m + \epsilon_D + \epsilon + (1-\lambda)G^2)\sum_{r=1}^{\tau-1} C_r + \eta G^2 + \eta^2\gamma_m G\tau C_2
\end{aligned} \tag{60}$$

and,

$$\begin{aligned}
C_1 &= \frac{G + (\tau-1)B\theta_m}{1-\delta_2} \\
C_r &= \frac{2G + (r-1)B\theta_m}{1-\delta_2} \\
C_2 &= \frac{2G + (\tau-1)B\theta_m}{1-\delta_2},
\end{aligned} \tag{61}$$

$\epsilon_D > 0$ and $\epsilon > 0$ are upper bounds for the approximation errors of the Hessian matrix.

*Proof.* According to the smoothness condition of $F(\mathbf{x})$. We have

$$\mathbb{E}[F(\mathbf{x}_{t+\tau+1}) - F(\mathbf{x}^*)] \leq \mathbb{E}[F(\mathbf{v}_{t+\tau}) - F(\mathbf{x}^*)] + \mathbb{E}[\langle \nabla F(\mathbf{v}_{t+\tau}), (\mathbf{x}_{t+\tau+1} - \mathbf{v}_{t+\tau})\rangle] + \frac{\gamma_m}{2}\mathbb{E}[\|\mathbf{x}_{t+\tau+1} - \mathbf{v}_{t+\tau}\|^2] \tag{62}$$

Based on the update law, we can obtain

$$\mathbb{E}[F(\mathbf{x}_{t+\tau+1}) - F(\mathbf{x}^*)]$$

$$\leq \mathbb{E}[F(\mathbf{v}_{t+\tau}) - F^*] - \eta\mathbb{E}[\langle \nabla F(\mathbf{v}_{t+\tau}), \mathbf{g}(\mathbf{x}_{t+\tau})\rangle] - \eta\mathbb{E}[\langle \nabla F(\mathbf{v}_{t+\tau}), \sum_{r=0}^{\tau-1} W'\mathbf{g}^{dc,r}(\mathbf{x}_t)\rangle]$$

$$+ \frac{\gamma_m \eta^2}{2}\mathbb{E}[\|\mathbf{g}(\mathbf{x}_{t+\tau}) + \sum_{r=0}^{\tau-1} W'\mathbf{g}^{dc,r}(\mathbf{x}_t)\|^2]$$

$$\leq \mathbb{E}[F(\mathbf{v}_{t+\tau}) - F^*] - \eta\mathbb{E}[\langle \nabla F(\mathbf{v}_{t+\tau}), \mathbf{g}(\mathbf{x}_{t+\tau})\rangle] - \eta\mathbb{E}[\langle \nabla F(\mathbf{v}_{t+\tau}), \tau\nabla F(\mathbf{v}_{t+\tau})\rangle] \tag{63}$$

$$+ \eta\mathbb{E}[\langle \nabla F(\mathbf{v}_{t+\tau}), \tau\nabla F(\mathbf{v}_{t+\tau}) - \sum_{r=0}^{\tau-1} \nabla F(\mathbf{v}_{t+r})\rangle] + \eta\mathbb{E}[\langle \nabla F(\mathbf{v}_{t+\tau}), \sum_{r=0}^{\tau-1} \nabla F(\mathbf{v}_{t+r}) - \mathcal{F}^h(\mathbf{x}_t)\rangle]$$

$$+ \eta\mathbb{E}[\langle \nabla F(\mathbf{v}_{t+\tau}), \mathbb{E}[\mathcal{G}^h] - \sum_{r=0}^{\tau-1} W'\mathbf{g}^{dc,r}(\mathbf{x}_t)\rangle] + \frac{\gamma_m \eta^2}{2}\mathbb{E}[\|\mathbf{g}(\mathbf{x}_{t+\tau}) + \sum_{r=0}^{\tau-1} W'\mathbf{g}^{dc,r}(\mathbf{x}_t)\|^2]$$

We next investigate each term on the right hand side. Based on Lemma 6, we can obtain

$$F(\mathbf{x}_{t+\tau}) \geq F(\mathbf{v}_{t+\tau}) - 2G\eta\frac{G + (\tau-1)B\theta_m}{1 - \delta_2} \tag{64}$$

such that

$$F(\mathbf{x}_{t+\tau}) - F^* \geq F(\mathbf{v}_{t+\tau}) - F^* - 2G\eta\frac{G + (\tau-1)B\theta_m}{1 - \delta_2} \tag{65}$$

For the term $-\eta\mathbb{E}[\langle \nabla F(\mathbf{v}_{t+\tau}), g(\mathbf{x}_{t+\tau})\rangle]$, we can quickly get that is is bounded above by $\eta G^2$ due to the Cauthy-Schwarz inequality. Then for term $-\eta\mathbb{E}[\langle \nabla F(\mathbf{v}_{t+\tau}), \tau\nabla F(\mathbf{v}_{t+\tau})\rangle]$, one can get the following relationship due to the PL condition.

$$-\eta\mathbb{E}[\langle \nabla F(\mathbf{v}_{t+\tau}), \tau\nabla F(\mathbf{v}_{t+\tau})\rangle] \leq -2\eta\tau\mu(F(\mathbf{v}_{t+\tau}) - F^*) \tag{66}$$

Combining $F(\mathbf{v}_{t+\tau}) - F^*$, we have

$$(1 - 2\eta\tau\mu)(F(\mathbf{v}_{t+\tau}) - F^*)$$

$$\leq (1 - 2\eta\tau\mu)[(F(\mathbf{x}_{t+\tau}) - F^*) + 2G\eta\frac{G + (\tau-1)B\theta_m}{1 - \delta_2}] \tag{67}$$

Based on Lemma 4, we have known that

$$\|\nabla F(\mathbf{v}_{t+r}) - \nabla\mathcal{F}^{h,r}(\mathbf{x}_t)\| \leq \frac{\xi_m}{2}\eta^2[\frac{2G + (r-1)B\theta_m}{1 - \delta_2}]^2; \tag{68}$$

for $r \geq 1$, while for $r = 0$, it can be obtained that

$$\|\nabla F(\mathbf{v}_t) - \nabla F(\mathbf{x}_t)\| \leq 2\gamma_m\eta\frac{G + (\tau-1)B\theta_m}{1 - \delta_2}. \tag{69}$$

Since

$$\eta\mathbb{E}[\langle \nabla F(\mathbf{v}_{t+\tau}), \sum_{r=0}^{\tau-1} \nabla F(\mathbf{v}_{t+r}) - \mathcal{F}^h(\mathbf{x}_t)\rangle] \leq \eta\mathbb{E}[\|\nabla F(\mathbf{v}_{t+\tau})\|\|\sum_{r=0}^{\tau-1} \nabla F(\mathbf{v}_{t+r}) - \mathcal{F}^h(\mathbf{x}_t)\|]$$

$$\leq \mathbb{E}[\|\nabla F(\mathbf{v}_{t+\tau})\|\sum_{r=0}^{\tau-1} \|\nabla F(\mathbf{v}_{t+r}) - \mathcal{F}^h(\mathbf{x}_t)\|] \tag{70}$$

The first inequality follows from Cauthy-Schwarz inequality and the second one follows from Triangle inequality. Hence, we can have

$$\eta\mathbb{E}[\langle\nabla F(\mathbf{v}_{t+\tau}), \sum_{r=0}^{\tau-1}\nabla F(\mathbf{v}_{t+r}) - \mathcal{F}^h(\mathbf{x}_t)\rangle] \leq \frac{\eta^3\xi_m G}{2(1-\delta_2)}\sum_{r=1}^{\tau-1}[2G + B(r-1)\theta_m] + 2\eta^2 G\gamma_m\frac{G + (\tau-1)B\theta_m}{1-\delta_2}$$

(71)

According to Lemma 4, the following relationship can be obtained,

$$\mathbb{E}[\langle\nabla F(\mathbf{v}_{t+\tau}), \mathbb{E}[\mathcal{G}^h(\mathbf{x}_t)] - \sum_{r=0}^{\tau-1}W'\mathbf{g}^{dc,r}(\mathbf{x}_t)\rangle] \leq \frac{\eta^2 G}{1-\delta_2}(\gamma_m + \epsilon_D + \epsilon + (1-\lambda)G^2)\sum_{r=1}^{\tau-1}[2G + (r-1)B\theta_m] + G\eta\tau\sigma$$

(72)

The last term is $\eta\mathbb{E}[\langle\nabla F(\mathbf{v}_{t+\tau}), \tau\nabla F(\mathbf{v}_{t+\tau}) - \sum_{r=0}^{\tau-1}\nabla F(\mathbf{v}_{t+r})\rangle]$, which can be rewritten such that

$$\eta\mathbb{E}[\langle\nabla F(\mathbf{v}_{t+\tau}), \tau\nabla F(\mathbf{v}_{t+\tau}) - \sum_{r=0}^{\tau-1}\nabla F(\mathbf{v}_{t+r})\rangle]$$
$$\leq\eta\mathbb{E}[\|\nabla F(\mathbf{v}_{t+\tau})\|\|\nabla F(\mathbf{v}_{t+\tau}) - \nabla F(\mathbf{v}_t) + \cdots + \nabla F(\mathbf{v}_{t+\tau}) - \nabla F(\mathbf{v}_{t+\tau-1})\|]$$
$$\leq\eta\mathbb{E}[\|\nabla F(\mathbf{v}_{t+\tau})\|\|\nabla F(\mathbf{v}_{t+\tau}) - \nabla F(\mathbf{v}_t)\| + \cdots + \|\nabla F(\mathbf{v}_{t+\tau}) - \nabla F(\mathbf{x}_{t+\tau-1})\|]$$

(73)

Using the smoothness condition, we then can bound the term by deriving the following relationship with Lemma 1 and Lemma 3,

$$\eta\mathbb{E}[\langle\nabla F(\mathbf{v}_{t+\tau}), \tau\nabla F(\mathbf{v}_{t+\tau}) - \sum_{r=0}^{\tau-1}\nabla F(\mathbf{v}_{t+r})\rangle] \leq \eta^2\gamma_m G\tau\frac{2G + (\tau-1)B\theta_m}{1-\delta_2}$$

(74)

We combine the upper bounds of each term on the right hand side to produce the following relationship.

$$\mathbb{E}[F(\mathbf{x}_{t+\tau+1}) - F(\mathbf{x}^*)] \leq (1 - 2\eta\mu\tau)(F(\mathbf{x}_{t+\tau}) - F^*) + 2(1 - 2\eta\mu\tau)G\eta\frac{G + (\tau-1)B\theta_m}{1-\delta_2}$$
$$+ \frac{\eta^3\xi_m G}{2(1-\delta_2)}\sum_{r=1}^{\tau-1}[2G + (r-1)B\theta_m] + 2\eta^2 G\gamma_m\frac{G + (\tau-1)B\theta_m}{1-\delta_2} + G\eta\tau\sigma + \eta G^2$$
$$+ \frac{\eta^2 G}{1-\delta_2}(\gamma_m + \epsilon_D + \epsilon + (1-\lambda)G^2)\sum_{r=1}^{\tau-1}[2G + (r-1)B\theta_m] + \eta^2\gamma_m G\tau\frac{2G + (\tau-1)B\theta_m}{1-\delta_2}.$$

(75)

We now know that

$$\mathbb{E}[F(\mathbf{x}_{t+1}) - F^*] \leq (1 - 2\eta\tau\mu)\mathbb{E}[F(\mathbf{x}_t) - F^*] + Q,$$

(76)

subtracting the constant $\frac{Q}{2\mu\tau\eta}$ from both sides, one obtains

$$\mathbb{E}[F(\mathbf{x}_{t+1}) - F^*] - \frac{Q}{2\mu\tau\eta} \leq (1 - 2\eta\mu\tau)\mathbb{E}[F(\mathbf{x}_t) - F^*] + Q - \frac{Q}{2\mu\tau\eta}$$
$$= (1 - 2\eta\mu\tau)(\mathbb{E}[F(\mathbf{x}_t) - F^*] - \frac{Q}{2\mu\tau\eta})$$

(77)

Observe that the above inequality is a contraction inequality since $0 < 2\eta\mu\tau \leq 1$ due to $0 < \eta \leq \frac{1}{2\mu\tau}$. The result thus follows by applying the inequality repeatedly through iteration $t \in \mathbb{N}$. □

Another scenario that could be of interest is the strongly convex objective. As Theorem 1 has shown with a properly set constant step size, PC-ASGD is able to converge to the neighborhood of the optimal solution

with a linear rate. This also applies to the strongly convex objective in which the strong convexity implies the PL condition, while the constants are subject to changes. We now proceed to give the proof for the generally convex case.

**Theorem 2**: Let Assumptions 1, 2 and 3 hold. Assume that the delay compensated gradients are uniformly bounded, i.e., there exists a scalar $B > 0$ such that for all $T \geq 1$

$$\|\mathbf{g}^{dc,r}(\mathbf{x}_t)\| \leq B, \quad \forall t \geq 0 \ and \ 0 \leq r \leq \tau - 1, \tag{78}$$

and there exists $C > 0$,

$$\mathbb{E}[\|\mathbf{x}_t - \mathbf{x}^*\|] \leq C, \tag{79}$$

where $\mathbf{x}^* \in \mathrm{argmin} F(\mathbf{x})$. Then for the iterations generated by PC-ASGD, there exists $0 < \eta < \frac{1}{20\gamma_m}$, such that

$$\mathbb{E}[F(\bar{\mathbf{x}}_T) - F^*] \leq \frac{\|\mathbf{x}_1 - \mathbf{x}^*\|^2}{T\eta} + \frac{A}{\eta}, \tag{80}$$

where $A = 10\eta^2\sigma_*^2 + 10\eta^2\sigma^2 + 20\eta^4 G^2 C_1^2 + 5\eta^2\theta_m^2\tau^2 B^2 + 2\eta C\theta_m\tau B + 2G\eta^2 C_1(2C+1), C_1 = \frac{G + (\tau-1)B\theta_m}{1 - \delta_2}, \sigma_*^2 := \mathbb{E}\|\mathbf{g}(\mathbf{x}^*) - \nabla F(\mathbf{x}^*)\|^2, \bar{\mathbf{x}}_T := \frac{1}{T}\sum_{t=1}^{T}\mathbf{x}_t$.

*Proof.* According to the compact update law, we have

$$\|\mathbf{x}_{t+\tau+1} - \mathbf{x}^*\|^2 = \|\mathcal{W}_{t+\tau}\mathbf{x}_{t+\tau} - \eta(\mathbf{g}(\mathbf{x}_{t+\tau}) + \theta_{t+\tau}\sum_{r=0}^{\tau-1} W'\mathbf{g}^{dc,r}(\mathbf{x}_t)) - \mathbf{x}^*\|^2. \tag{81}$$

As $\mathbf{v}_{t+\tau} = \mathcal{W}_{t+\tau}\mathbf{x}_{t+\tau}$, we can obtain

$$\|\mathbf{x}_{t+\tau+1} - \mathbf{x}^*\|^2 = \|\mathbf{v}_{t+\tau} - \mathbf{x}^*\|^2 - 2\eta\langle\mathbf{v}_{t+\tau} - \mathbf{x}^*, \mathbf{g}(\mathbf{x}_{t+\tau}) + \theta_{t+\tau}\sum_{r=0}^{\tau-1} W'\mathbf{g}^{dc,r}(\mathbf{x}_t)\rangle$$
$$+ \eta^2\|\mathbf{g}(\mathbf{x}_{t+\tau}) + \theta_{t+\tau}\sum_{r=0}^{\tau-1} W'\mathbf{g}^{dc,r}(\mathbf{x}_t)\|^2. \tag{82}$$

For convenience, we define that $\Gamma_{t+\tau} = \mathbf{g}(\mathbf{x}_{t+\tau}) + \theta_{t+\tau}\sum_{r=0}^{\tau-1} W'\mathbf{g}^{dc,r}(\mathbf{x}_t)$. Hence, the above equation can be rewritten as

$$\|\mathbf{x}_{t+\tau+1} - \mathbf{x}^*\|^2 = \|\mathbf{v}_{t+\tau} - \mathbf{x}^*\|^2 + \eta^2\|\Gamma_{t+\tau}\|^2$$
$$+ 2\eta\langle\mathbf{x}^* - \mathbf{v}_{t+\tau}, \mathbf{g}(\mathbf{v}_{t+\tau})\rangle + 2\eta\langle\mathbf{x}^* - \mathbf{v}_{t+\tau}, \mathbf{g}(\mathbf{x}_{t+\tau}) - \mathbf{g}(\mathbf{v}_{t+\tau})\rangle$$
$$+ 2\eta\langle\mathbf{x}^* - \mathbf{v}_{t+\tau}, \theta_{t+\tau}\sum_{r=0}^{\tau-1} W'\mathbf{g}^{dc,r}(\mathbf{x}_t)\rangle. \tag{83}$$

Taking expectation on both sides leads to the following relationship:

$$\mathbb{E}[\|\mathbf{x}_{t+\tau+1} - \mathbf{x}^*\|^2] \leq \mathbb{E}[\|\mathbf{x}_{t+\tau} - \mathbf{x}^*\|^2] + \eta^2\mathbb{E}[\|\Gamma_{t+\tau}\|^2]$$
$$+ 2\eta\mathbb{E}[\langle\mathbf{x}^* - \mathbf{v}_{t+\tau}, \nabla F(\mathbf{v}_{t+\tau})\rangle] + 2\eta\mathbb{E}[\langle\mathbf{x}^* - \mathbf{v}_{t+\tau}, \nabla F(\mathbf{x}_{t+\tau}) - \nabla F(\mathbf{v}_{t+\tau})\rangle]$$
$$+ 2\eta\mathbb{E}[\langle\mathbf{x}^* - \mathbf{v}_{t+\tau}, \theta_{t+\tau}\sum_{r=0}^{\tau-1} W'\mathbf{g}^{dc,r}(\mathbf{x}_t)\rangle]. \tag{84}$$

The inequality holds due to the basic property for the projection Sundhar Ram et al. (2010). For the last two terms on the right hand side of the above inequality, we can leverage Cauchy-Schwartz inequality to obtain the upper bounds. For $2\eta\mathbb{E}[\langle\mathbf{x}^* - \mathbf{v}_{t+\tau}, \nabla F(\mathbf{v}_{t+\tau})\rangle]$, we will use Lemma 2 to reformulate. We next investigate $\eta^2\mathbb{E}[\|\Gamma_{t+\tau}\|^2]$. Before that, we introduce a theoretical fact for the generally convex smooth functions.

**Variance transfer: gradient noise** (Lemma 4.20) in Garrigos & Gower (2023). If $F$ is smooth and convex, then for all $\mathbf{x}$ we have that

$$\mathbb{E}[\|\mathbf{g}(\mathbf{x})\|^2] \leq 4\gamma_m(F(\mathbf{x}) - F^*) + 2\sigma_*^2, \tag{85}$$

where $\mathbf{g}(\mathbf{x})$ is the stochastic gradient, $\sigma_*^2$ is the variance of stochastic gradient at $\mathbf{x}^*$. Rewrite $\|\Gamma_{t+\tau}\|^2 = \|\mathbf{g}(\mathbf{x}_{t+\tau}) + \nabla F(\mathbf{x}_{t+\tau}) - \nabla F(\mathbf{x}_{t+\tau}) + \mathbf{g}(\mathbf{v}_{t+\tau}) - \mathbf{g}(\mathbf{v}_{t+\tau}) + \nabla F(\mathbf{v}_{t+\tau}) - \nabla F(\mathbf{v}_{t+\tau}) + \theta_{t+\tau} \sum_{r=0}^{\tau-1} W' \mathbf{g}^{dc,r}(\mathbf{x}_t)\|^2$. We then have the following relationship:

$$
\begin{aligned}
\mathbb{E}[\|\mathbf{x}_{t+\tau+1} - \mathbf{x}^*\|^2] \leq\; & \mathbb{E}[\|\mathbf{x}_{t+\tau} - \mathbf{x}^*\|^2] + 5\eta^2 \mathbb{E}[\|\mathbf{g}(\mathbf{v}_{t+\tau})\|^2] + 5\eta^2 \mathbb{E}[\|\mathbf{g}(\mathbf{x}_{t+\tau}) - \nabla F(\mathbf{x}_{t+\tau})\|^2] \\
& + 5\eta^2 \mathbb{E}[\|\mathbf{g}(\mathbf{v}_{t+\tau}) - \nabla F(\mathbf{v}_{t+\tau})\|^2] + 5\eta^2 \mathbb{E}[\|\nabla F(\mathbf{x}_{t+\tau}) - \nabla F(\mathbf{v}_{t+\tau})\|^2] \\
& + 5\eta^2 \mathbb{E}[\|\theta_{t+\tau} \sum_{r=0}^{\tau-1} W' \mathbf{g}^{dc,r}(\mathbf{x}_t)\|^2] + 2\eta \mathbb{E}[F^* - F(\mathbf{v}_{t+\tau})] \\
& + 2\eta \mathbb{E}[\|\mathbf{x}^* - \mathbf{v}_{t+\tau}\| \|\nabla F(\mathbf{x}_{t+\tau}) - \nabla F(\mathbf{v}_{t+\tau})\|] + 2\eta \mathbb{E}[\|\mathbf{x}^* - \mathbf{v}_{t+\tau}\| \|\theta_{t+\tau} \sum_{r=0}^{\tau-1} W' \mathbf{g}^{dc,r}(\mathbf{x}_t)\|].
\end{aligned}
\tag{86}
$$

The last inequality holds due to the basic inequality $\|\sum_{i=1}^N \mathbf{a}_i\|^2 \leq N \sum_{i=1}^N \|\mathbf{a}_i\|^2$, the convexity property, and Cauchy-Schwartz inequality. By substituting Eq. 85 into Eq. 86, the following relationship can be obtained

$$
\begin{aligned}
\mathbb{E}[\|\mathbf{x}_{t+\tau+1} - \mathbf{x}^*\|^2] \leq\; & \mathbb{E}[\|\mathbf{x}_{t+\tau} - \mathbf{x}^*\|^2] + 20\eta^2 \gamma_m \mathbb{E}[F(\mathbf{v}_{t+\tau}) - F^*] + 10\eta^2 \sigma_*^2 \\
& + 5\eta^2 \mathbb{E}[\|\mathbf{g}(\mathbf{x}_{t+\tau}) - \nabla F(\mathbf{x}_{t+\tau})\|^2] \\
& + 5\eta^2 \mathbb{E}[\|\mathbf{g}(\mathbf{v}_{t+\tau}) - \nabla F(\mathbf{v}_{t+\tau})\|^2] + 5\eta^2 \mathbb{E}[\|\nabla F(\mathbf{x}_{t+\tau}) - \nabla F(\mathbf{v}_{t+\tau})\|^2] \\
& + 5\eta^2 \mathbb{E}[\|\theta_{t+\tau} \sum_{r=0}^{\tau-1} W' \mathbf{g}^{dc,r}(\mathbf{x}_t)\|^2] + 2\eta \mathbb{E}[F^* - F(\mathbf{v}_{t+\tau})] \\
& + 2\eta \mathbb{E}[\|\mathbf{x}^* - \mathbf{v}_{t+\tau}\| \|\nabla F(\mathbf{x}_{t+\tau}) - \nabla F(\mathbf{v}_{t+\tau})\|] + 2\eta \mathbb{E}[\|\mathbf{x}^* - \mathbf{v}_{t+\tau}\| \|\theta_{t+\tau} \sum_{r=0}^{\tau-1} W' \mathbf{g}^{dc,r}(\mathbf{x}_t)\|] \\
\leq\; & \mathbb{E}[\|\mathbf{x}_{t+\tau} - \mathbf{x}^*\|^2] + 2\eta(10\gamma_m \eta - 1)\mathbb{E}[F(\mathbf{x}_{t+\tau}) - F^*] + 10\eta^2 \sigma_*^2 \\
& + 10\eta^2 \sigma^2 + 20\eta^4 G^2 C_1^2 + 5\eta^2 \theta_m^2 \tau^2 B^2 + 2\eta C(2G\eta C_1 + \theta_m \tau B).
\end{aligned}
\tag{87}
$$

The second inequality follows from Assumption 3, Eq. 47 and bounds for the predicted gradients. With mathematical manipulation, the above inequality can be written as

$$
\begin{aligned}
2\eta(1 - 10\gamma_m \eta)\mathbb{E}[F(\mathbf{v}_{t+\tau}) - F^*] \leq\; & \mathbb{E}[\|\mathbf{x}_{t+\tau} - \mathbf{x}^*\|^2] - \mathbb{E}[\|\mathbf{x}_{t+\tau+1} - \mathbf{x}^*\|^2] \\
& + 10\eta^2 \sigma_*^2 + 10\eta^2 \sigma^2 + 20\eta^4 G^2 C_1^2 + 5\eta^2 \theta_m^2 \tau^2 B^2 + 2\eta C(2G\eta C_1 + \theta_m \tau B)
\end{aligned}
\tag{88}
$$

Due to $\eta \leq \frac{1}{20\gamma_m}$, $1 - 10\gamma_m \eta \geq \frac{1}{2}$ such that $\eta \mathbb{E}[F(\mathbf{v}_{t+\tau}) - F^*] \leq 2\eta(1 - 10\gamma_m \eta)\mathbb{E}[F(\mathbf{v}_{t+\tau}) - F^*]$. Dividing both sides of Eq. 88 by $\eta$ yields the following

$$
\begin{aligned}
\mathbb{E}[F(\mathbf{v}_{t+\tau}) - F^*] \leq\; & \frac{1}{\eta}(\mathbb{E}[\|\mathbf{x}_{t+\tau} - \mathbf{x}^*\|^2] - \mathbb{E}[\|\mathbf{x}_{t+\tau+1} - \mathbf{x}^*\|^2]) \\
& + \frac{1}{\eta}(10\eta^2 \sigma_*^2 + 10\eta^2 \sigma^2 + 20\eta^4 G^2 C_1^2 + 5\eta^2 \theta_m^2 \tau^2 B^2 + 2\eta C(2G\eta C_1 + \theta_m \tau B)).
\end{aligned}
\tag{89}
$$

Similar to Lemma 6, we can obtain that $F(\mathbf{v}_{t+\tau}) \geq F(\mathbf{x}_{t+\tau}) - 2G\eta \frac{G+(\tau-1)B\theta_m}{1-\delta_2}$. Then it is immediately obtained that $F(\mathbf{v}_{t+\tau}) - F^* \geq F(\mathbf{x}_{t+\tau}) - F^* - 2G\eta \frac{G+(\tau-1)B\theta_m}{1-\delta_2}$. With this, the following relationship can be obtained

$$
\mathbb{E}[F(\mathbf{x}_{t+\tau}) - F^*] \leq \frac{1}{\eta}(\mathbb{E}[\|\mathbf{x}_{t+\tau} - \mathbf{x}^*\|^2] - \mathbb{E}[\|\mathbf{x}_{t+\tau+1} - \mathbf{x}^*\|^2]) + \frac{A}{\eta},
\tag{90}
$$

where $A = 10\eta^2 \sigma_*^2 + 10\eta^2 \sigma^2 + 20\eta^4 G^2 C_1^2 + 5\eta^2 \theta_m^2 \tau^2 B^2 + 2\eta C \theta_m \tau B + 2G\eta^2 C_1(2C+1)$. Recursively summing over $t$ from 1 to $T$ and replacing $t + \tau$ with $t$ grants us the following relationship

$$
\sum_{t=1}^T \mathbb{E}[F(\mathbf{x}_t) - F^*] \leq \frac{\|\mathbf{x}_1 - \mathbf{x}^*\|^2}{\eta} + \frac{AT}{\eta}.
\tag{91}
$$

Dividing both sides by $T$ in the last relationship attains the following

$$\frac{1}{T}\sum_{t=1}^{T}\mathbb{E}[F(\mathbf{x}_t) - F^*] \leq \frac{\|\mathbf{x}_1 - \mathbf{x}^*\|^2}{T\eta} + \frac{A}{\eta}. \tag{92}$$

Using that $F$ is convex with Jensen inequality gives the desirable result. □

In the sequel, we provide the details for the smooth nonconvex functions.

**Theorem 3**: Let Assumptions 1,2 and 3 hold. Assume that the delay compensated gradients are uniformly bounded, i.e., there exists a scalar $B > 0$ such that

$$\|\mathbf{g}^{dc,r}(\mathbf{x}_t)\| \leq B, \quad \forall t \geq 0 \text{ and } 0 \leq r \leq \tau - 1, \tag{93}$$

and that

$$\mathbb{E}[\|\mathbf{g}^{dc}(\mathbf{x}_t)\|^2] \leq M. \tag{94}$$

Then for the iterates generated by PC-ASGD, there exists $0 < \eta < \frac{1}{\gamma_m}$, such that for all $T \geq 1$,

$$\frac{1}{T}\sum_{t=1}^{T}\mathbb{E}[\|\nabla F(\mathbf{x}_t)\|^2] \leq \frac{2(F(\mathbf{x}_1) - F^*)}{T\eta} + \frac{R}{\eta}, \tag{95}$$

where

$$R = 2G\eta^2 C_1 + \frac{\tau\eta^2\gamma_m M}{2} + \frac{\eta\sigma^2}{2} + \eta\sigma\tau B + 2\eta^2\gamma_m(\tau B + G)C_1.$$

*Proof.* According to the smoothness condition of $F(\mathbf{x})$, we have

$$F(\mathbf{x}_{t+\tau+1}) - F(\mathbf{v}_{t+\tau})$$

$$\leq \langle\nabla F(\mathbf{v}_{t+\tau}), \mathbf{x}_{t+\tau+1} - \mathbf{v}_{t+\tau}\rangle + \frac{\gamma_m}{2} + \|\mathbf{x}_{t+\tau+1} - \mathbf{v}_{t+\tau}\|^2$$

$$= \langle\nabla F(\mathbf{v}_{t+\tau}), -\eta(\sum_{r=0}^{\tau-1}W'\mathbf{g}^{dc,r}(\mathbf{x}_t) + \mathbf{g}(\mathbf{x}_{t+\tau}))\rangle + \frac{\eta^2\gamma_m}{2}\|\sum_{r=0}^{\tau-1}W'\mathbf{g}^{dc,r} + \mathbf{g}(\mathbf{x}_{t+\tau})\|^2$$

$$= \langle\nabla F(\mathbf{v}_{t+\tau}) - \nabla F(\mathbf{x}_{t+\tau}) + \nabla F(\mathbf{x}_{t+\tau}), \eta(\sum_{r=0}^{\tau-1}W'\mathbf{g}^{dc,r}(\mathbf{x}_t) + \mathbf{g}(\mathbf{x}_{t+\tau}))\rangle + \frac{\eta^2\gamma_m}{2}\|\sum_{r=0}^{\tau-1}W'\mathbf{g}^{dc,r} + \mathbf{g}(\mathbf{x}_{t+\tau})\|^2$$

$$= -\eta\langle\nabla F(\mathbf{x}_{t+\tau}), \sum_{r=0}^{\tau-1}W'\mathbf{g}^{dc,r}(\mathbf{x}_t) + \mathbf{g}(\mathbf{x}_{t+\tau})\rangle + \eta\langle(\nabla F(\mathbf{v}_{t+\tau}) - \nabla F(\mathbf{x}_{t+\tau}), \sum_{r=0}^{\tau-1}W'\mathbf{g}^{dc,r}(\mathbf{x}_t) + \mathbf{g}(\mathbf{x}_{t+\tau}))\rangle$$

$$+ \frac{\eta^2\gamma_m}{2}\|\sum_{r=0}^{\tau-1}W'\mathbf{g}^{dc,r} + \mathbf{g}(\mathbf{x}_{t+\tau})\|^2$$

$$= -\frac{\eta}{2}[\|\nabla F(\mathbf{x}_{t+\tau})\|^2 + \|\sum_{r=0}^{\tau-1}W'\mathbf{g}^{dc,r}(\mathbf{x}_t) + \mathbf{g}(\mathbf{x}_{t+\tau})\|^2 - \|\nabla F(\mathbf{x}_{t+\tau}) - (\sum_{r=0}^{\tau-1}W'\mathbf{g}^{dc,r}(\mathbf{x}_t) + \mathbf{g}(\mathbf{x}_{t+\tau}))\|^2]$$

$$+ \eta\langle\nabla F(\mathbf{x}_{t+\tau}) - \nabla F(\mathbf{v}_{t+\tau}), \sum_{r=0}^{\tau-1}W'\mathbf{g}^{dc,r}(\mathbf{x}_t) + \mathbf{g}(\mathbf{x}_{t+\tau})\rangle + \frac{\eta^2\gamma_m}{2}\|\sum_{r=0}^{\tau-1}W'\mathbf{g}^{dc,r} + \mathbf{g}(\mathbf{x}_{t+\tau})\|^2$$

$$= -\frac{\eta}{2}\|\nabla F(\mathbf{x}_{t+\tau})\|^2 - \frac{\eta}{2}\|\sum_{r=0}^{\tau-1}W'\mathbf{g}^{dc,r}(\mathbf{x}_t) + \mathbf{g}(\mathbf{x}_{t+\tau})\|^2 + \frac{\eta}{2}(\|\nabla F(\mathbf{x}_{t+\tau}) - \mathbf{g}(\mathbf{x}_{t+\tau})\|^2 + \|\sum_{r=0}^{\tau-1}W'\mathbf{g}^{dc,r}(\mathbf{x}_t)\|^2$$

$$- 2\langle\nabla F(\mathbf{x}_{t+\tau}) - \mathbf{g}(\mathbf{x}_{t+\tau}), \sum_{r=0}^{\tau-1}W'\mathbf{g}^{dc,r}(\mathbf{x}_t)\rangle) + \eta\langle\nabla F(\mathbf{x}_{t+\tau}) - \nabla F(\mathbf{v}_{t+\tau}), \sum_{r=0}^{\tau-1}W'\mathbf{g}^{dc,r}(\mathbf{x}_t) + \mathbf{g}(\mathbf{x}_{t+\tau})\rangle$$

$$+ \frac{\eta^2 \gamma_m}{2} \| \sum_{r=0}^{\tau-1} W' \mathbf{g}^{dc,r} + \mathbf{g}(\mathbf{x}_{t+\tau}) \|^2$$

$$= -\frac{\eta}{2} \|\nabla F(\mathbf{x}_{t+\tau})\|^2 - (\frac{\eta}{2} - \frac{\eta^2 \gamma_m}{2}) \| \sum_{r=0}^{\tau-1} W' \mathbf{g}^{dc,r}(\mathbf{x}_t) + \mathbf{g}(\mathbf{x}_{t+\tau}) \|^2 + \frac{\eta}{2} \|\nabla F(\mathbf{x}_{t+\tau}) - \mathbf{g}(\mathbf{x}_{t+\tau})\|^2 + \frac{\eta}{2} \| \sum_{r=1}^{\tau-1} W' \mathbf{g}^{dc,r}(\mathbf{x}_t) \|^2$$

$$- \eta \langle \nabla F(\mathbf{x}_{t+\tau}) - \mathbf{g}(\mathbf{x}_{t+\tau}), \sum_{r=1}^{\tau-1} W' \mathbf{g}^{dc,r}(\mathbf{x}_t) \rangle + \eta \langle \nabla F(\mathbf{x}_{t+\tau}) - \nabla F(\mathbf{v}_{t+\tau}), \sum_{r=0}^{\tau-1} W' \mathbf{g}^{dc,r}(\mathbf{x}_t) + \mathbf{g}(\mathbf{x}_{t+\tau}) \rangle$$

$$= -\frac{\eta}{2} \|\nabla F(\mathbf{x}_{t+\tau})\|^2 + (\frac{\eta^2 \gamma_m}{2} - \frac{\eta}{2}) \| \sum_{r=0}^{\tau-1} W' \mathbf{g}^{dc,r}(\mathbf{x}_t) \|^2 + (\frac{\eta^2 \gamma_m}{2} - \frac{\eta}{2}) \|\mathbf{g}(\mathbf{x}_{t+\tau})\|^2$$

$$+ (\frac{\eta^2 \gamma_m}{2} - \frac{\eta}{2}) \langle \mathbf{g}(\mathbf{x}_{t+\tau}), \sum_{r=0}^{\tau-1} W' \mathbf{g}^{dc,r}(\mathbf{x}_t) \rangle + \frac{\eta}{2} \|\nabla F(\mathbf{x}_{t+\tau}) - \mathbf{g}(\mathbf{x}_{t+\tau})\|^2 + \frac{\eta}{2} \| \sum_{r=1}^{\tau-1} W' \mathbf{g}^{dc,r}(\mathbf{x}_t) \|^2$$

$$- \eta \langle \nabla F(\mathbf{x}_{t+\tau}) - \mathbf{g}(\mathbf{x}_{t+\tau}), \sum_{r=1}^{\tau-1} W' \mathbf{g}^{dc,r}(\mathbf{x}_t) \rangle + \eta \langle \nabla F(\mathbf{x}_{t+\tau}) - \nabla F(\mathbf{v}_{t+\tau}), \sum_{r=0}^{\tau-1} W' \mathbf{g}^{dc,r}(\mathbf{x}_t) + \mathbf{g}(\mathbf{x}_{t+\tau}) \rangle$$

$$\leq -\frac{\eta}{2} \|\nabla F(\mathbf{x}_{t+\tau})\|^2 + (\frac{\eta^2 \gamma_m}{2} - \frac{\eta}{2}) \| \sum_{r=0}^{\tau-1} W' \mathbf{g}^{dc,r}(\mathbf{x}_t) \|^2 + (\frac{\eta^2 \gamma_m}{2} - \frac{\eta}{2}) \|\mathbf{g}(\mathbf{x}_{t+\tau})\|^2$$

$$+ (\frac{\eta^2 \gamma_m}{2} - \frac{\eta}{2}) \|\mathbf{g}(\mathbf{x}_{t+\tau})\| \| \sum_{r=0}^{\tau-1} W' \mathbf{g}^{dc,r}(\mathbf{x}_t) \| + \frac{\eta}{2} \|\nabla F(\mathbf{x}_{t+\tau}) - \mathbf{g}(\mathbf{x}_{t+\tau})\|^2 + \frac{\eta}{2} \| \sum_{r=1}^{\tau-1} W' \mathbf{g}^{dc,r}(\mathbf{x}_t) \|^2$$

$$+ \eta \|\nabla F(\mathbf{x}_{t+\tau}) - \mathbf{g}(\mathbf{x}_{t+\tau})\| \| \sum_{r=1}^{\tau-1} W' \mathbf{g}^{dc,r}(\mathbf{x}_t) \| + \eta \|\nabla F(\mathbf{x}_{t+\tau}) - \nabla F(\mathbf{v}_{t+\tau})\| \| \sum_{r=0}^{\tau-1} W' \mathbf{g}^{dc,r}(\mathbf{x}_t) + \mathbf{g}(\mathbf{x}_{t+\tau}) \|.$$

The first inequality follows from the smooth property of the objective. The last inequality follows from Cauthy-Schwarz inequality. The left hand side of the above inequality can be rewritten:

$$F(\mathbf{x}_{t+\tau+1}) - F(\mathbf{x}_{t+\tau}) + F(\mathbf{x}_{t+\tau}) - F(\mathbf{v}_{t+\tau})$$

Taking expectations for both sides, with the last inequality, we have

$$\mathbb{E}[F(\mathbf{x}_{t+\tau+1}) - F(\mathbf{x}_{t+\tau})]$$

$$\leq \mathbb{E}[F(\mathbf{v}_{t+\tau}) - F(\mathbf{x}_{t+\tau})] - \frac{\eta}{2} \mathbb{E}[\|\nabla F(\mathbf{x}_{t+\tau})\|^2] + \frac{\eta^2 \gamma_m - \eta}{2} \mathbb{E}[\| \sum_{r=0}^{\tau-1} W' \mathbf{g}^{dc,r}(\mathbf{x}_t) \|^2] + \frac{\eta^2 \gamma_m - \eta}{2} \mathbb{E}[\|\mathbf{g}(\mathbf{x}_{t+\tau})\|^2]$$

$$+ \frac{\eta^2 \gamma_m - \eta}{2} \mathbb{E}[\|\mathbf{g}(\mathbf{x}_{t+\tau})\| \| \sum_{r=0}^{\tau-1} W' \mathbf{g}^{dc,r}(\mathbf{x}_t) \|] + \frac{\eta}{2} \mathbb{E}[\|\nabla F(\mathbf{x}_{t+\tau}) - \mathbf{g}(\mathbf{x}_{t+\tau})\|^2] + \frac{\eta}{2} \mathbb{E}[\| \sum_{r=1}^{\tau-1} W' \mathbf{g}^{dc,r}(\mathbf{x}_t) \|^2]$$

$$+ \eta \mathbb{E}[\|\nabla F(\mathbf{x}_{t+\tau}) - \mathbf{g}(\mathbf{x}_{t+\tau})\| \| \sum_{r=1}^{\tau-1} W' \mathbf{g}^{dc,r}(\mathbf{x}_t) \|] + \eta \mathbb{E}[\|\nabla F(\mathbf{x}_{t+\tau}) - \nabla F(\mathbf{v}_{t+\tau})\| \| \sum_{r=0}^{\tau-1} W' \mathbf{g}^{dc,r}(\mathbf{x}_t) + \mathbf{g}(\mathbf{x}_{t+\tau}) \|]$$

$$\leq G \mathbb{E}[\|\mathbf{v}_{t+\tau} - \mathbf{x}_{t+\tau}\|] - \frac{\eta}{2} \mathbb{E}[\|\nabla F(\mathbf{x}_{t+\tau})\|^2] + \frac{\eta^2 \gamma_m - \eta}{2} \tau \sum_{r=0}^{\tau-1} \mathbb{E}[\|W' \mathbf{g}^{dc,r}(\mathbf{x}_t)\|^2] + \frac{\eta^2 \gamma_m - \eta}{2} \mathbb{E}[\|\mathbf{g}(\mathbf{x}_{t+\tau})\|^2]$$

$$+ \frac{\eta^2 \gamma_m - \eta}{2} \mathbb{E}[\|\mathbf{g}(\mathbf{x}_{t+\tau})\| \| \sum_{r=0}^{\tau-1} W' \mathbf{g}^{dc,r}(\mathbf{x}_t) \|] + \frac{\eta}{2} \mathbb{E}[\|\nabla F(\mathbf{x}_{t+\tau}) - \mathbf{g}(\mathbf{x}_{t+\tau})\|^2] + \frac{\eta}{2} \mathbb{E}[\| \sum_{r=1}^{\tau-1} W' \mathbf{g}^{dc,r}(\mathbf{x}_t) \|^2]$$

$$+ \eta \mathbb{E}[\|\nabla F(\mathbf{x}_{t+\tau}) - \mathbf{g}(\mathbf{x}_{t+\tau})\| \| \sum_{r=1}^{\tau-1} W' \mathbf{g}^{dc,r}(\mathbf{x}_t) \|] + \eta \mathbb{E}[\|\nabla F(\mathbf{x}_{t+\tau}) - \nabla F(\mathbf{v}_{t+\tau})\| \| \sum_{r=0}^{\tau-1} W' \mathbf{g}^{dc,r}(\mathbf{x}_t) + \mathbf{g}(\mathbf{x}_{t+\tau}) \|]$$

$$\leq -\frac{\eta}{2} \mathbb{E}[\|\nabla F(\mathbf{x}_{t+\tau})\|^2] + \frac{\tau^2 \eta^2 \gamma_m M}{2} + \frac{\eta \sigma^2}{2} + \eta \sigma \tau B + 2\eta^2 \gamma_m (\tau B + G + \frac{G}{\eta \gamma_m}) \frac{G + (\tau - 1) B \theta_m}{1 - \delta_2}$$

$$(96)$$

The last inequality follows from the smoothness condition of $F(\mathbf{x})$ and the bounded gradient, respectively, as well as $\eta < \frac{1}{\gamma_m}$. Hence, by replacing $t + \tau$ with $t$, one can obtain

$$\mathbb{E}[F(\mathbf{x}_{t+1}) - F(\mathbf{x}_t)] \leq -\frac{\eta}{2}\mathbb{E}[\|\nabla F(\mathbf{x}_t)\|^2] + R \tag{97}$$

where $R$ indicates the constant term on the right hand side of the inequality. As we assume that $F(\mathbf{x})$ is bounded from below, applying the last inequality from 1 to $T$, one can get

$$F^* - F(\mathbf{x}_1) \leq \mathbb{E}[F(\mathbf{x}_{t+1})] - F(\mathbf{x}_1) \leq -\frac{\eta}{2}\sum_{t=1}^{T}\mathbb{E}[\|\nabla F(\mathbf{x}_t)\|^2] + TR \tag{98}$$

which results in

$$\sum_{t=1}^{T}\mathbb{E}[\|\nabla F(\mathbf{x}_t)\|^2] \leq \frac{2[(F(\mathbf{x}_1) - F^*) + TR]}{\eta} \tag{99}$$

Dividing both sides by $T$, the desirable results are obtained. $\qquad\square$

## C   Detailed Settings of Deep Learning Models

**Model Settings** For the PreResNet110 (*model 1*), DenseNet (*model 2*), ResNet20 (*model 3*) and Efficient-Net (*model 4*), models' architectures are shown in He et al. (2016b), Huang et al. (2017), He et al. (2016a) and Tan & Le (2019) respectively. The batch size is selected as 128. After hyperparameter searching in $(0.1, 0.01, 0.001)$, the learning rate is set as 0.01 for the first 160 epochs and changed to 0.001. The decays are applied in epochs $(80, 120, 160, 200)$. The approximation coefficient $\lambda$ is set as 1. $\lambda = 0.001$ is first tried as suggested by DC-ASGD Zheng et al. (2017) and the results show that the predicting step doesn't affect the training process. By considering the upper bound of 1, a set of values $(0.001, 0.1, 1)$ are tried, and $\lambda = 1$ is applied according to the performance.

**Hardware environment.** Our experiments are implemented and evaluated at GTX-1080 ti with Intel Xenon 2.55GHz processor with 32GB RAM.

Table 5: Performance comparison in TinyImageNet and Time Series dataset

| Model & dataset | Pre110 TinyImageNet | DesNet TinyImageNet | EfficientNet TinyImageNet | LSTM Wind Turbine Data |
|---|---|---|---|---|
| PC-ASGD (Ours) | $\mathbf{58.0 \pm 1.4}$ | $\mathbf{61.4 \pm 0.7}$ | $\mathbf{74.8 \pm 0.9}$ | $\mathbf{71.2 \pm 0.5}$ |
| D-ASGD Lian et al. (2017) | $52.1 \pm 0.3$ | $57.5 \pm 0.2$ | $70.4 \pm 0.5$ | $66.2 \pm 0.1$ |
| DC-s3gd Rigazzi (2019) | $55.1 \pm 0.8$ | $58.5 \pm 1.4$ | $73.2 \pm 1.2$ | $61.4 \pm 1.1$ |
| D-ASGD with IS Du et al. (2020) | $53.2 \pm 0.9$ | $58.1 \pm 1.2$ | $73.4 \pm 0.7$ | $69.2 \pm 0.2$ |
| Adaptive Braking Venigalla et al. (2020) | $55.2 \pm 1.2$ | $60.2 \pm 1.1$ | $67.3 \pm 1.5$ | $66.5 \pm 1.2$ |

Table 6: Performance evaluation of ResNet20 on CIFAR-10

| | 20 agents | | | | | | |
|---|---|---|---|---|---|---|---|
| | PC-ASGD | | P-ASGD | | C-ASGD | | Baseline |
| Model & dataset | acc. (%) | o.p. (%) | acc. (%) | o.p. (%) | acc.(%) | o.p. (%) | acc. (%) |
| ResNet 20, CIFAR-10 | $\mathbf{84.9 \pm 0.6}$ | $\mathbf{2.4 \pm 0.7}$ | $82.9 \pm 0.7$ | $0.4 \pm 0.8$ | $83.8 \pm 0.8$ | $1.3 \pm 0.9$ | $82.5 \pm 0.1$ |

acc.–accuracy, o.p.–outperformed comparing to baseline.

# D   More Empirical Results with different datasets

We also adopt our numerical studies on TinyImageNet Le & Yang (2015) and Wind turbine data set Liu et al. (2014). For TinyImageNet, we adopt PreResNet110 He et al. (2016b), DenseNet Huang et al. (2017), and EfficientNet Tan & Le (2019). For the wind turbine data set, we use LSTM[3] in Lei et al. (2019) to classify the fault in the wind turbine.

Results in Tab. 5 shows the effectiveness of our proposed methods in different models, datasets, and even different tasks (time series classification). It further demonstrates the generality of our proposed framework.

We also supplement the experiment with ResNet20 on CIFAR-10 to ablate the functions of the P-step and C-step in Tab. 6. The quantitative results are consistent with Tab. 2, showing the benefits of our PC steps design.

---

[3]Actually, we use SGD-based optimizer for better analysis instead of Adam in Lei et al. (2019), hence we do not achieve the best results in Lei et al. (2019). But our framework shows the best performances among other framework handling delay.

