# OpenReview forum: "Asynchronous Training Schemes in Distributed Learning with Time Delay"
_TMLR — Accepted by TMLR_

### Review · Reviewer_N9HH · 2023-12-03

**Summary Of Contributions:**

The paper presents a new delay-tolerant algorithm for decentralized learning referred to as Predicting Clipping Asynchronous Stochastic Gradient Descent (PC-ASGD). PC-ASGD includes two steps - the predicting step leverages the gradient prediction using Taylor expansion to reduce the staleness of the outdated weights while the clipping step selectively drops the outdated weights to alleviate their negative effects. The authors also present a practical variant of PC-ASGD. Convergence analysis of PC-ASGD for weakly strongly convex functions
and nonconvex functions is shown. Experimental results validate the efficiency of the proposed method.

**Audience:**

Yes

**Claims And Evidence:**

Yes

**Requested Changes:**

1. Page 13 contains one figure and the rest is just empty space. The paper needs reformatting.
2. In equation 2, the LHS is $g_k^{dc,r}$ but $r$ is fixed from 0 to $\tau-1$. Correct the notation.
3. The clipping step can be replaced by the first term of the predicting step.
4. Add discussion on the algorithmic differences between DCS3GD and PC-ASGD especially in terms of the delay compensation used (the predicting step).
5. Experiments on optimized models meant for CIFAR datasets such as ResNet-20 could be useful. The current results are presented on over-parameterized networks.

**Strengths And Weaknesses:**

Strengths:
1. Proposed a novel algorithm that includes weighted delay compensation for asynchronous decentralized learning.
2. Related work section gives a good overview of all the existing works.
3. Theoretical guarantees for the convergence rate of PC-ASGD are presented.
4. Experimental results are presented for CIFAR-10 and CIFAR-100 for two different model architectures and 3 different network sizes.
5. Compared with relevant baselines
6. Presents ablation study on parameter $\theta$, delay time $\tau$, and network size $N$.

Weaknesses/Questions:
1. The second term in the predicting step needs access to $x_t^k$ and $g_{t-\tau}^k$. Does this mean the proposed method need $2\times$ communication for unreliable clients as they have to send both model parameters and gradients? Also, shouldn't the predicting step be $x_{t+1, pre}^i = \sum_{j \in R} w_{ij} x_t^j - \eta g_i(x_t^i) + \sum_{k \in R^c}x_{t-\tau}^k-\eta g_k^{dc}(x_{t-\tau}^k)$?
2. According to the given delay compensation step, we have $x_{t+2}^k = x_t^k - \eta [g_k(x_t^k)+\lambda g_k(x_t^k) \odot g_k(x_t^k) \odot (x_{t+1}^i - x_t^i)]$. It is not clear to me how predicting agent k's gradient $g_k(x_{t+1}^k)$ using the agent i's model differences is an estimate for $(x_{t+1}^k - x_t^k)$. I see that since we have no access to $x_{t+1}^k$, we replaced $(x_{t+1}^k - x_t^k)$ in the Taylor series expansion with $(x_{t+1}^i - x_t^i)$. Can you explain why this is reasonable?
3. The clipping step is equivalent to $x_{t+1, clip}^i = \sum_{j \in R} w_{ij} x_t^j - \eta g_i(x_t^i)$. Introducing $\tilde{w}_{ij}$ is unnecessary.
4. Assumption 2c requires the mixing matrix to be positive semi-definite. This is not a standard assumption in decentralized learning algorithms. Why does PC-ASGD need this assumption?
5. Table 4 shows that PC-ASGD-PV gives the best performance. In this experiment, can you provide some insights on how many times $\theta$ takes the value of 1 vs 0 for PC-ASGD-PV? Let's say the model is trained for T iterations using PC-ASGD-PV. Then for a given agent-$i$, the clipping step was activated for $c_i$ iteration and the predicting step was activated for $p_i$ iterations. what are the values of $\frac{1}{N} \sum_j \frac{c_j}{T}$ and $\frac{1}{N} \sum_j \frac{p_j}{T}$?
6. PreResNet110 and DenseNet are over-parameterized networks for CIFAR networks. Do you get similar improvements in accuracy over baseline if compact networks such as ResNet-20 are used?
7. Section 5.2 says "The detailed model structures are illustrated in the Appendix". I couldn't find any details on model architectures in the appendix.
8. The experiments are only conducted on a fully connected network. Will the proposed algorithm work for sparser decentralized topologies such as Torus and Social Network?
9. What are the communication and compute overheads of PC-ASGD as compared to D-ASGD?

---

> ### Comment · Reviewer_N9HH · 2024-01-10
> **Regarding 2x communication**
>
> The communication part is still not clear to me. The updated paper mentions that "However, the term $x_t^k - \eta g_k^{dc}(x_{t-\tau}^k)$ in the predicting step is essentially calculated by agent k first and then communicated to agent i". But from Equation 2,  $g_k^{dc}(x_{t-\tau}^k)$ is the delay compensated gradient estimated at agent i and uses $(x_{t-\tau+r}^i-x_{t-\tau}^i)$ for r ranging from $0$ to $\tau-1$. How can agent k compute $g_k^{dc}(x_{t-\tau}^k)$ without having access to $x_{t-\tau}^i, x_{t-\tau+1}^i, ..., x_{t-1}^i$?

---

> ### Comment · Reviewer_N9HH · 2024-01-11
> **Reply by Reviewer N9HH**
>
> Clarifications:
> 1. PC-ASGD in the tables and figures is an implementation of Algorithm 2: PC-ASGD-PV, correct?
> 2. In Figure 4, the P and C choice stabilizes indicating that $\theta$ is taking a constant value. For a given agent, is this value proportional to the time delay? For example, if the time delay is set to 20 iterations and each epoch contains 50 iterations then is the $\theta \sim \frac{20}{50}$. It would be great if you could give some insights on the correlation of $\theta$ to the time delay value along with the dependence on the number of unreliable clients.
> 3.  Do all the unreliable clients send the information at the same time? In particular, is their delay synchronized?
> 4. A13 hypothetically claims that  $(x_{t}^i-x_{t-\tau}^k)$ may cause larger error bound compared to $(x_{t}^i-x_{t-\tau}^i)$ for the computation of $g_k^{dc}$. Do you have any empirical results to back up this statement?
>
> Minor comments:
>
> 1. Please add a limitation section to discuss compute (latency), communication, and memory overheads. The proposed algorithm has compute overhead reflected as latency (shown in Fig 5), communication overhead (2x for unreliable clients), and memory overhead of storing  $x_{t-\tau}^i, x_{t-\tau+1}^i, ..., x_{t-1}^i$ to compute delay compensation as compared to  D-ASGD. It is also important to acknowledge any new hyperparameters the algorithm introduces that need tuning.
> 2. Table 3's caption seems to be incorrect.

---

> ### Comment · Reviewer_N9HH · 2024-01-12
> **Reply by Reviewer N9HH**
>
> Thanks for the feedback on my comments.
>
> Regarding Q4: Comparing tables 2 and 3 shows that  using $(x_{t}^i-x_{t-\tau}^k)$ for prediction performs better than $(x_{t}^i-x_{t-\tau}^i)$ in 2/4 cases and almost same in one case. At least empirically, it looks like utilizing $(x_{t}^i-x_{t-\tau}^k)$ performs better for the prediction step.

---

### Review · Reviewer_pyta · 2023-12-24

**Summary Of Contributions:**

1. The introduction of PC-ASGD offers a novel solution to the persistent issue of stale weights in distributed systems due to communication delays. The proposed two-step approach, involving gradient prediction and clipping, is a clever strategy that adeptly balances the need for efficiency and accuracy in learning.

2. By effectively addressing the challenges posed by variable delays, PC-ASGD significantly improves the accuracy of distributed learning systems compared with other async schemes.

3. The authors provide a comprehensive theoretical analysis, confirming the algorithm's soundness, supplemented by empirical evidence from experiments with benchmark datasets.

**Audience:**

Yes

**Claims And Evidence:**

Yes

**Requested Changes:**

I think this paper presents a simple yet effective algorithm for Asynchronous Federated Learning with robust theoretical backing. However, it would be beneficial for the authors to expand their experimental section.

1. The paper currently assumes a uniform network delay for all agents in a connected graph, which may not align with the variability seen in real-world networks. In practical scenarios, communication delays can differ and change over time among agents. Therefore, it would be beneficial for the authors to conduct additional experiments that simulate time-varying and agent-specific communication delays. This would provide a more comprehensive and realistic assessment of the algorithm's performance under diverse and dynamic network conditions.

2. The evaluation of the proposed method would benefit from incorporating a broader range of datasets and models. Additionally, expanding the scale of the network-connected agents would be crucial to thoroughly validate the algorithm's effectiveness in more extensive and varied network environments.

3. The hardware environment used in the experiments should be involved in the paper.

**Strengths And Weaknesses:**

Strengths:

1. Simple yet effective solution: The paper presents PC-ASGD as a straightforward yet highly effective solution to the challenges in distributed learning, which involves predicting and clipping gradients to alleviate the affect of communication delay from neighbor agents. It outperforms the existing methods in both accuracy and efficiency.

2. Strong Theoretical Foundation: The paper provides a robust theoretical analysis of the algorithm, including a detailed examination of its convergence rate, which adds to the credibility of this proposed algorithm.


Weaknesses:

1. The paper's experimental section might be limited in its comparative analysis for different scenarios, such as in network settings with varying delays or in much larger scales.

---

### Review · Reviewer_7PTK · 2023-12-29

**Summary Of Contributions:**

The main contributions are the following:

1. The paper addresses the challenge of stale weights or gradients in distributed optimization problems and introduces a novel algorithm, PC-ASGD, to mitigate these issues. Notably, it adopts a unified framework, considering both synchronous and asynchronous algorithms.

2. PC-ASGD strategically balances the projection and clipping steps, enabling convergence for both smooth nonconvex problems and those with the PL condition. Under the bounded delay assumption, accounting for decentralized computation and gradient compensation, the paper establishes PC-ASGD as the first provable algorithm in this setting.

**Audience:**

Yes

**Claims And Evidence:**

Yes

**Requested Changes:**

It should be “AGP” in the tenth-to-last row on page 2.

**Strengths And Weaknesses:**

Strengths:

1. Novel Algorithm: A new algorithm, PC-ASGD, is proposed to address the challenge of stale weights and gradients in distributed optimization. Section 3.1 also provides a clear description of the idea behind the algorithm.

2. Comprehensive Experiments: Lots of experiments are conducted, effectively illustrating the effectiveness of the newly proposed algorithms.

Weakness:

1. I am curious about the convergence rate in the convex setting. If PC-ASGD can match the state-of-the-art in the convex setting, it would make it even more impressive and highlight its versatility as a unified framework.

2. For Section 5.8, testing one more example that satisfies the PL condition would be beneficial to further validate the convergence results.

---

### Decision · Action_Editor_LBw5 · 2024-02-14

**Recommendation:** Accept as is

**Comment:**

The paper introduces PC-ASGD, a novel algorithm addressing the challenge of stale weights or gradients in distributed optimization problems. PC-ASGD employs a two-step approach involving gradient prediction and clipping to mitigate the effects of communication delays, offering a unified framework for both synchronous and asynchronous algorithms. Theoretical analysis confirms the soundness and convergence rate of PC-ASGD, while extensive experiments showcase its effectiveness in various scenarios, model architectures, and network sizes, supplemented by an ablation study. PC-ASGD outperforms existing methods in terms of accuracy and efficiency, supported by comparisons with relevant baselines. The paper also provides a comprehensive overview of related work and addresses potential questions regarding its application in different network topologies and the comparison of communication and compute overheads with other algorithms. The paper is well-written well, and the authors have addressed the comments that reviewers had and incorporated the suggested changes into the updated version.

**Audience:**

Yes, the submission would be of interest to ML researchers, in general, and in particular, researchers focusing on distributed optimization and learning would find the paper's findings relevant and valuable. The novel algorithm, PC-ASGD, addresses a common challenge in distributed optimization problems, making it of interest to researchers seeking new approaches to improve optimization efficiency and accuracy in distributed systems. ML practitioners would benefit from the submissions as the proposed algorithm offers a practical solution to mitigate the effects of communication delays in distributed learning systems, potentially improving the performance of machine learning models deployed in real-world distributed environments. Finally, academic researchers and students studying machine learning and optimization techniques would likely be interested in the theoretical analysis and empirical evidence presented in the paper. The clear description of the algorithm, along with its theoretical guarantees and experimental results, can serve as valuable learning material and inspire future research in the field.

**Claims And Evidence:**

Yes, the claims made in the submission are supported by accurate, convincing, and clear evidence. The paper introduces a novel algorithm, PC-ASGD, to address the challenge of stale weights or gradients in distributed optimization problems. The evidence supporting the claims includes (a) theoretical analysis for both convex and nonconvex functions; (b) extensive experiments illustrating the effectiveness of PC-ASGD for various scenarios, model architectures, and network sizes; (c) comparisons with existing methods in terms of accuracy and efficiency and (d) a comprehensive overview of related work. Overall, the evidence presented in the submission supports the claims made regarding the effectiveness and novelty of PC-ASGD as a solution to distributed optimization challenges.